# DISENTANGLED REPRESENTATION LEARNING FOR PARAMETRIC PARTIAL DIFFERENTIAL EQUATIONS

## ABSTRACT

Neural operators (NOs) have demonstrated remarkable success in learning mappings between function spaces, serving as efficient approximators for the forward solutions of complex physical systems governed by partial differential equations (PDEs). However, while effective as black-box solvers, they offer limited insight into the underlying physical mechanism, due to the lack of interpretable representations of the physical parameters that drive the system. To tackle this challenge, we propose a new paradigm for learning disentangled representations from neural operator parameters, thereby effectively solving an inverse problem. Specifically, we introduce DisentangO, a novel hyper-neural operator architecture designed to unveil and disentangle the latent physical factors of variation embedded within the black-box neural operator parameters. At the core of DisentangO is a multi-task neural operator architecture that distills the varying parameters of the governing PDE through a task-wise adaptive layer, coupled with a variational autoencoder that disentangles these variations into identifiable latent factors. By learning these disentangled representations, our model not only enhances physical interpretability but also enables more robust generalization across diverse physical systems. Empirical evaluations across supervised, semi-supervised, and unsupervised learning contexts show that DisentangO effectively extracts meaningful and interpretable latent features, bridging the gap between predictive performance and physical understanding in neural operator frameworks.

## 1 INTRODUCTION

Interpretability in machine learning (ML) refers to the ability to understand and explain how models make decisions (Rudin et al., 2022; Molnar, 2020). As ML systems grow more complex, especially with the use of deep learning and ensemble methods (Sagi & Rokach, 2018), the reasoning behind their predictions can become opaque. Interpretability addresses this challenge by making the models more transparent, enabling users to trust the outcomes, detect biases, and identify potential flaws. It is a critical factor in applying AI responsibly, especially in fields where accountability and fairness are essential (Cooper et al., 2022). In physics, where models endeavor to capture the governing laws and physical principles, understanding how a model arrives at its predictions is vital for verifying that it aligns with known scientific theories. Interpretable models also allow researchers to gain insights into the learned relationships, ensuring that the predictions are physically meaningful and consistent with real-world behaviors. This transparency is key for advancing scientific discovery, validating results, and enhancing trust in models used for complex physical systems.

Discovering interpretable representations of physical parameters in learning physical systems is challenging due to the intricate nature of real-world phenomena and the often implicit relationships between variables. In physics, parameters like force, energy, and velocity are governed by well-established laws, and extracting these parameters in a way that aligns with physical intuition requires models that go beyond mere pattern recognition. Traditional ML models may fit the data but fail to represent these parameters in a physically interpretable way. To address this, recent developments include the integration of physical constraints into the learning process (Raissi et al., 2019), such as embedding conservation laws (Liu et al., 2023; 2024a), symmetries (Mattheakis et al., 2019), or invariances (Cohen & Welling, 2016) directly into model architectures. Additionally, methods like symbolic regression (Biggio et al., 2021) and sparse modeling (Carroll et al., 2009) aim to discover simple, interpretable expressions that capture the underlying dynamics. However, balancing model

complexity with interpretability remains a significant challenge, as overly complex models may obscure the true physical relationships, while oversimplified models risk losing critical details of the system's behavior.

In this work, we introduce **DisentangO**, a novel variational hyper-neural operator architecture to disentangle physical factors of variation from black-box neural operator parameters for solving parametric PDEs. Neural operators (NOs) (Li et al., 2020a;c) learn mappings between infinite-dimensional function spaces in the form of integral operators, making them powerful tools for discovering continuum physical laws by manifesting the mappings between spatial and/or spatiotemporal data; see You et al. (2022a); Liu et al. (2024a;b; 2023); Ong et al. (2022); Cao (2021); Lu et al. (2019; 2021); Goswami et al. (2022); Gupta et al. (2021) and references therein. However, most NOs play the role of efficient forward surrogates for the underlying physical system under a supervised learning setting. As a result, they act as black-box universal approximators for a single physical system governed by a fixed set of PDE parameters, and lack interpretability with respect to the underlying physical laws. In contrast, the key innovation of DisentangO lies in its use of a hypernetwork architecture that distills the varying physical parameters of the governing PDE from multiple physical systems through an unsupervised learning setting. The distilled variations are further disentangled into distinct physical factors to enhance physical understanding and promote robust generalization. Consequently, DisentangO effectively extracts meaningful and interpretable physical features, thereby simultaneously solving both the forward and inverse problems. **Our key contributions** are:

- We bridge the divide between predictive accuracy and physical interpretability, and introduce a new paradigm that simultaneously performs physics modeling (i.e., as a forward PDE solver) and governing physical mechanism discovery (i.e., as an inverse PDE solver).
- We propose a novel variational hyper-neural operator architecture, which we coin DisentangO. DisentangO extracts the key physical factors of variation from black-box neural operator parameters of multiple physical systems. These factors are then disentangled into distinct latent factors that enhance physical interpretation and robust generalization.
- We provide theoretical analysis on the component-wise identifiability of the true generative factors in physics modeling: by learning from multiple physical systems, the variability of hidden physical states in these systems promotes identifiability.
- We explore the practical utility of disentanglement rather than pursuing it as an end goal, and perform experiments across a broad range of settings including supervised, semi-supervised, and unsupervised learning. Our results show that DisentangO effectively extracts meaningful and interpretable physical features for enhanced interpretability.

## 2 BACKGROUND AND RELATED WORK

**Neural operators.** Learning complex physical systems directly from data is crucial in many scientific and engineering applications (Ghaboussi et al., 1998; Liu et al., 2024c; Ghaboussi et al., 1991; Carleo et al., 2019; Karniadakis et al., 2021; Zhang et al., 2018; Cai et al., 2022; Pfau et al., 2020; He et al., 2021; Jafarzadeh et al., 2024). Often, the governing laws of these systems are unknown and must be inferred from data through learning models that are both *resolution-invariant* to ensure consistent performance across various levels of discretizations and *interpretable* to domain experts. Neural operators (NOs) address the first aim by learning mappings between infinite-dimensional function spaces (Li et al., 2020a;b;c; You et al., 2022a; Ong et al., 2022; Cao, 2021; Lu et al., 2019; 2021; Goswami et al., 2022; Gupta et al., 2021). By capturing the relationship between spatial and spatio-temporal data, NOs offer a promising approach to discovering continuum physical surrogates, enabling more accurate and consistent predictions of the underlying responses of complex systems. Nevertheless, NOs lack an interpretable representation of the physical states driving the system, and as a result offer limited insight into the underlying physical mechanisms.

**Hypernetworks.** Hypernetworks (Ha et al., 2016; Schmidhuber, 1992; Stanley et al., 2009; Chauhan et al., 2024) are a class of neural network architectures that use one network to generate weights for another neural network. Then, both networks can be trained in an end-to-end differentiable manner. As such, hypernetworks allow soft weight sharing across different tasks, which makes them useful for transfer learning and dynamic information sharing (Chauhan et al., 2023). Hypernetworks can also be employed as a versatile technique in existing neural network architectures. For instance, in Nguyen et al.; Oh & Peng (2022), a hypernetwork is employed to generate parameters for a VAE model and enable multi-task learning. Similarly, Lee et al. (2023) integrates

the hypernetwork architecture with neural operators. However, none of the existing work discusses the capability of hypernetworks in hidden physics discovery from data.

**Forward and inverse PDE learning.** Existing NOs focus on providing efficient surrogates for the underlying physical systems as forward PDE solvers. They are often employed as black-box universal approximators but lack interpretability of the underlying physical principles. On the other hand, several deep learning methods have been developed as inverse PDE solvers (Fan & Ying, 2023; Molinaro et al., 2023; Jiang et al., 2022; Chen et al., 2023), aiming to reconstruct the PDE parameters from solution data. However, the inverse problem is typically more challenging due to its ill-posedness. To tackle this pathology, many NOs incorporate prior information via governing PDEs (Yang et al., 2021; Li et al., 2021), regularizers (Dittmer et al., 2020; Obmann et al., 2020; Ding et al., 2022; Chen et al., 2023), or additional operator structures (Uhlmann, 2009; Lai et al., 2019; Yilmaz, 2001). However, all these work rely on prior information on the operator form, which is not realistic in many real-world problems where neither the parameter field nor the model form are available. Worth noting is a recent work that leverages the attention mechanism to build a nonlinear kernel map in NO (Yu et al., 2024), with which a data-dependent kernel can be constructed to characterize the inverse mapping from data to the hidden governing mechanism. While effective, this approach does not disentangle the learned kernel or extract physically interpretable parameters. To the best of our knowledge, our proposed DisentangO is the first that tackles both forward (i.e., physics prediction) and inverse (i.e., physics discovery) PDE learning problems, while simultaneously inferring the governing distinct physical parameters from the learned NO.

**Disentangled representation learning.** Disentangled representation learning seeks to separate data into distinct, interpretable factors, where each factor captures an independent aspect of variation in the input. This area has critical implications for transfer learning, generative modeling, and fairness in AI, as it allows models to generalize across tasks by leveraging isolated features, making them more interpretable and modular. Disentangled representation learning has seen significant advances, primarily driven by models like $\beta$-VAE (Higgins et al., 2017), FactorVAE (Kim & Mnih, 2018), and InfoGAN (Chen et al., 2016), who encourage disentanglement by controlling the trade-off between reconstruction accuracy and latent regularization, or introduce mutual information constraints to encourage the discovery of interpretable latent factors. Early work in disentanglement largely focus on supervised or semi-supervised approaches (Ridgeway & Mozer, 2018; Shu et al., 2019; Mathieu et al., 2019), requiring labeled data for factors of variation. However, more recent advancements have concentrated on unsupervised learning methods (Duan et al., 2019), where disentanglement is achieved without explicit labels. Despite these efforts, fully unsupervised disentanglement remains challenging (Locatello et al., 2019), with ongoing research focusing on improving metrics, robustness, and applicability to more complex and high-dimensional data. In this context, although a large body of research has been done in disentanglement, its applications remain largely confined to fields like computer vision and robotics, where the latent dimensions typically correspond to human-interpretable (often visually recognizable) features, with very limited exploration in areas such as physical system learning (cf. Lingsch et al. (2024); Tong et al. (2024); Fotiadis et al. (2023) in supervised learning). Moreover, existing work focuses on disentangling from data, whereas our work is the first attempt that explores disentanglement from black-box neural network parameters.

## 3 DISENTANGO

We consider a series of complex systems with different hidden physical parameters:

$$\mathcal{K}_{\boldsymbol{b}}[\boldsymbol{u}](\boldsymbol{x}) = \boldsymbol{f}(\boldsymbol{x}), \quad \boldsymbol{x} \in \Omega. \tag{3.1}$$

Here, $\Omega \subset \mathbb{R}^s$ is the domain of interest, $\boldsymbol{f}(\boldsymbol{x})$ is a function representing the loading on $\Omega$, and $\boldsymbol{u}(\boldsymbol{x})$ is the corresponding solution of this system. $\mathcal{K}_{\boldsymbol{b}}$ represents the unknown governing law, e.g., balance laws, which is determined by the (possibly unknown and high-dimensional) parameter field $\boldsymbol{b}$. For instance, in a material modeling problem, $\mathcal{K}_{\boldsymbol{b}}$ often stands for the constitutive law and $\boldsymbol{b}$ can be a vector ($\boldsymbol{b} \in \mathbb{R}^{d_b}$) representing the homogenized material parameter field or a vector-valued function ($\boldsymbol{b} \in L^\infty(\Omega; \mathbb{R}^{d_b})$) representing the heterogeneous material properties. Both scenarios are considered in our empirical experiments in Section 4.

Many physical modeling tasks can be formulated as either a forward or an inverse PDE-solving problem. In a forward problem setting, one aims to find the PDE solution when given PDE information, including coefficient functions, boundary conditions, initial conditions, and loading sources. That means, given the governing operators $\mathcal{K}$, the parameter (field) $\boldsymbol{b}$, and loading field $\boldsymbol{f}(\boldsymbol{x})$ in equation 3.1, the goal is to solve for the corresponding solution field $\boldsymbol{u}(\boldsymbol{x})$, through classical PDE solvers (Brenner & Scott, 2007) or data-driven approaches (Lu et al., 2019; Li et al., 2020c). As a

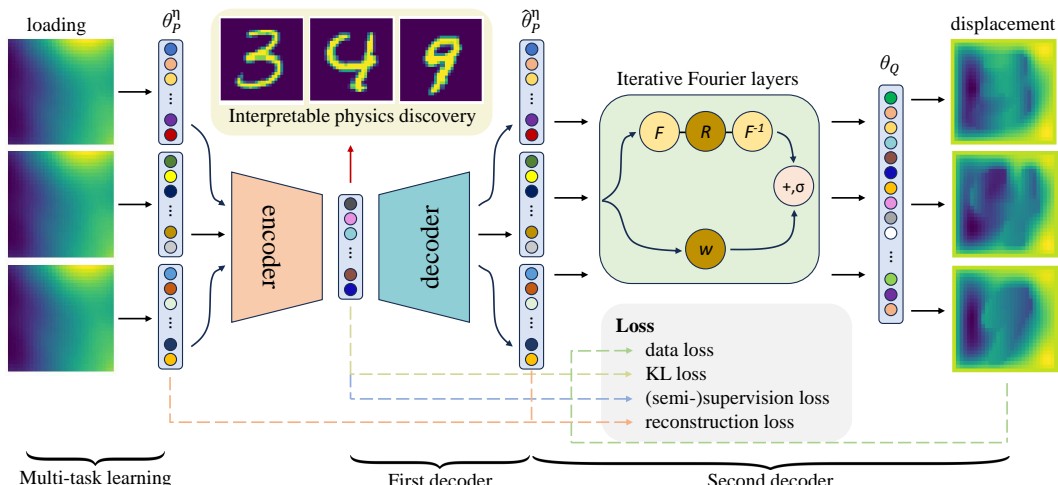

Figure 1: Illustration of DisentangO's architecture. The encoder disentangles meaningful physical factors from multiple physical systems, whereas the first and second decoders reconstruct the NO lifting-layer parameters $\theta_P^\eta$ and the forward map $\mathcal{G}$, respectively. Black arrows indicate forward computations, red arrow denotes where interpretable physics discovery occurs, and dashed arrows mark the sources of loss function computation.

result, a forward map is constructed:

$$\mathcal{G} : (\boldsymbol{b}, \boldsymbol{f}) \to \boldsymbol{u} . \tag{3.2}$$

Here, $\boldsymbol{b}$ and $\boldsymbol{f}$ are input vectors/functions, and $\boldsymbol{u}$ is the output function.

Conversely, solving an inverse PDE problem involves reconstructing the underlying full or partial PDE information from PDE solutions, where one seeks to construct an inverse map:

$$\mathcal{H} : (\boldsymbol{u}, \boldsymbol{f}) \to \boldsymbol{b} . \tag{3.3}$$

Unfortunately, solving an inverse problem is typically more challenging due to the ill-posed nature of the PDE model. In general, a small number of function pairs $(\boldsymbol{u}, \boldsymbol{f})$ from a single system does not suffice in inferring the underlying parameter field $\boldsymbol{b}$, making the inverse problem generally non-identifiable (Molinaro et al., 2023). Herein, we propose a novel neural architecture to alleviate the curse of ill-posedness without requiring prior knowledge. The key ingredients are: 1) the construction of a multi-task NO architecture, which solves both the forward (physics prediction) and inverse (physics discovery) problems simultaneously from multiple systems, and 2) a generative model to disentangle the features from NO parameters that contain critical information of $\boldsymbol{b}$.

### 3.1 VARIATIONAL HYPER-NEURAL OPERATOR AS A MULTI-TASK SOLVER

Formally, we consider $S$ training datasets from different systems, each of which contains $n_{train}$ function pairs $(\boldsymbol{u}, \boldsymbol{f}) \in \mathcal{U} \times \mathcal{F}$:

$$\mathcal{D}_{\mathrm{tr}} = \{\{(\boldsymbol{u}_i^\eta(\boldsymbol{x}), \boldsymbol{f}_i^\eta(\boldsymbol{x}))\}_{i=1}^{n_{train}}\}_{\eta=1}^S . \tag{3.4}$$

The index $\eta$ corresponds to different hidden PDE parameter field/set $\boldsymbol{b}^\eta \in \mathcal{B}$. Here, $\mathcal{U} \subset L^2(\Omega; \mathbb{R}^{d_u})$, $\mathcal{F} \subset L^2(\Omega; \mathbb{R}^{d_f})$ represent the (infinite-dimensional) Banach spaces of the solution function $\boldsymbol{u}$ and loading function $\boldsymbol{f}$, respectively, and $\mathcal{B}$ is a finite-dimensional manifold where the hidden parameter $\boldsymbol{b}$ takes values on. We further assume that the data satisfies the following generating process:

$$\boldsymbol{b} \sim \mathbb{P}_b , \ \boldsymbol{z} \sim p(\boldsymbol{z}|\boldsymbol{b}) , \tag{3.5}$$

where $\boldsymbol{z} \in \mathcal{Z} \subset \mathbb{R}^{d_z}$ is the latent embedding of $\boldsymbol{b}$. Then, we assume for each $\boldsymbol{b}^\eta$ that there exists an underlying parametric PDE system of equation 3.1. Each function pair $(\boldsymbol{u}_i^\eta, \boldsymbol{f}_i^\eta)$ satisfies:

$$\boldsymbol{u}|\boldsymbol{f} = \mathcal{K}_{\boldsymbol{z}}^{-1}[\boldsymbol{f}] , \ \boldsymbol{f} \sim \mathbb{P}_f , \tag{3.6}$$

where $\mathcal{K}_{\boldsymbol{z}}^{-1}$ is the ground-truth solution operator of the parametric PDE system of equation 3.1 with a given set of parameters $\boldsymbol{b}$.

Our first objective is to construct a surrogate operator of $\mathcal{K}_{\boldsymbol{z}}^{-1} : \mathcal{F} \to \mathcal{U}$, as the forward PDE solver for each system:

$$\mathcal{G}[\boldsymbol{f}_i^\eta; \theta_i^\eta](\boldsymbol{x}) \approx \boldsymbol{u}_i^\eta(\boldsymbol{x}) , \tag{3.7}$$

where $\theta_i^\eta$ denotes the corresponding trainable parameter set. Then, as the second objective, we construct an inverse map by discovering the hidden parameter field/set from $\theta_i^\eta$:

$$\mathcal{H}(\theta_i^\eta; \Theta_H) \approx \tilde{\boldsymbol{z}}_i^\eta . \tag{3.8}$$

Here, $\Theta_H$ is the trainable parameters of $\mathcal{H}$, and $\tilde{z}_i^\eta \in \mathbb{R}^{d_z}$ denotes the learned latent variables, which can be transformed to the ground-truth latent variable $z_i^\eta$ via an invertible function $h$. The goal of disentanglement is not to discover $b^\eta$, since some features may not be identifiable. For instance, in a Dirichlet boundary condition problem, $u_i^\eta(x) = 0$ for all $x \in \partial\Omega$, making $b^\eta$ not learnable on $\partial\Omega$. Hence, a more realistic goal is to discover the underlying mechanism of $b$ in the space of identifiability, i.e., $z$, or its equivalent transformation $\tilde{z}$, together with the solution operator $\mathcal{K}_z^{-1}$.

Note that in the first objective of equation 3.7, one is interested in finding the solution operator $\mathcal{G}^\eta$ under a supervised setting, in the form of function-to-function mappings, for all (hidden) parameters $b^\eta$ in the range of interest; while in the second objective of equation 3.8, the goal is to construct a vector-to-vector mapping from $\theta^\eta$ to $\tilde{z}^\eta$, which is unsupervised due to the hidden physics nature. We propose to employ variational autoencoder (VAE) (Kingma & Welling, 2013) as the representation learning approach for the second objective, and a meta-learned NO architecture as a universal solution operator for the first objective. The key is to pair these architectures as a hyper-neural operator, which simultaneously solves both forward and inverse modeling problems.

We now define the learning architecture. The overall objective is to maximize the log data likelihood:

$$\max E(\log(p(\theta, \mathcal{D}))) = \max \left[ E(\log(p(u|f, \theta) + \log(p(\theta|f)) + \log(p(f)) \right].$$

Note that $p(f)$ remains constant over different $\theta$, and the assumption in eq. 3.6 yields $\log(p(\theta|f)) = \log(p(\theta))$. Additionally, the assumption in eq. 3.5 guarantees that $\theta$ is generated from the latent space over $z$, hence $\log(p(\theta)) = \log \int_z p(\theta|z)p(z)dz$. The overall objective then becomes:

$$\max E(\log(p(\theta, \mathcal{D}))) = \max \left[ E(\log(p(u|f, \theta))) + E\left( \log \int_z p(\theta|z)p(z)dz \right) \right]. \tag{3.9}$$

However, the second term in this formulation is generally intractable. We use a variational posterior $q(\theta|z)$ to approximate the actual posterior $p(\theta|z)$ and maximize the evidence lower bound (ELBO):

$$L_{ELBO} = \frac{1}{S} \sum_{\eta=1}^S \left[ E_{q(z^\eta|\theta^\eta)} \log p(\theta^\eta|z^\eta) - D_{KL}\left( q(z^\eta|\theta^\eta) || p(z^\eta) \right) \right],$$

with $D_{KL}(\cdot||\cdot)$ denoting the KL divergence between two distributions. Putting everything together, we obtain the loss functional:

$$L_{loss} = \frac{1}{S} \sum_{\eta=1}^S \left[ -E(\log(p(u|f, \theta^\eta))) - E_{q(z^\eta|\theta^\eta)} \log p(\theta^\eta|z^\eta) + D_{KL}\left( q(z^\eta|\theta^\eta) || p(z^\eta) \right) \right]. \tag{3.10}$$

The above formulation naturally lends itself to a hierarchical variational autoencoder (HVAE) architecture (Vahdat & Kautz, 2020). The encoder aims to obtain the posterior $q_{\mu_z, \Sigma_z}(z^\eta|\theta_P^\eta)$, which estimates the inverse map $\mathcal{H}$:

$$\hat{z} \sim q_{\mu_z, \Sigma_z}(\hat{z}^\eta|\theta_P^\eta). \tag{3.11}$$

Then, the first decoder $\hat{g}$ processes the estimated latent variable $\hat{z}$ and reconstructs the corresponding NO parameter $\hat{\theta}_P$:

$$\hat{\theta}_P = \hat{g}(\hat{z}). \tag{3.12}$$

Lastly, the second decoder reconstructs the forward map $\mathcal{G}$, by further taking the estimated $\hat{\theta}_P$ and the loading function $f$ and estimating the output function $u$:

$$\hat{u} = \hat{\mathcal{G}}[f; \hat{\theta}_P]. \tag{3.13}$$

For this purpose, we employ the implicit Fourier neural operator (IFNO)[*], which is a provably universal approximator for PDE solution operators with a relatively small number of trainable parameters (You et al., 2022b). To provide a universal $\hat{\mathcal{G}}$ for different domains, a multi-task learning strategy is adopted following MetaNO (Zhang et al., 2023), which applies task-wise adaptation only to the first layer of NOs. Specifically, an $L$-layer multi-task IFNO has the following form:

$$\mathcal{G}[f; \theta^\eta](x) = \mathcal{G}[f; \theta_P^\eta, \theta_J, \theta_Q](x) := \mathcal{Q}_{\theta_Q} \circ (\mathcal{J}_{\theta_J})^L \circ \mathcal{P}_{\theta_P^\eta}[f](x), \tag{3.14}$$

where $\mathcal{P}, \mathcal{Q}$ are shallow-layer MLPs that map a low-dimensional vector to a high-dimensional vector and vice versa, parameterized by $\theta_Q$ and $\theta_P^\eta$, respectively. Each intermediate layer, $\mathcal{J}$, is constructed as a mimetic of a fixed-point iteration step and parameterized by $\theta_J$. Supported by the universal approximator analysis (Zhang et al., 2023), different PDEs share common iterative ($\theta_J$) and projection ($\theta_Q$) parameters, with all the information about parameter $b$ embedded in the task-wise lifting parameters $\theta_P^\eta$. Hence, the encoder and first decoder structure only need to encode and reconstruct $\theta_P^\eta$, since all other NO parameters are invariant to the change of $z$. This construction substantially reduces the degree of freedoms in $\theta^\eta$, making the invertibility assumption in the next section feasible. As suggested by MetaNO, we train one neural network for each task, i.e., assuming $\theta_i^\eta = \theta^\eta$. Then, the lifting layer parameter $\theta_P^\eta$ is taken as the task-wise parameter[†] where $z^\eta$ is estimated from.

---

[*] We point out that the proposed strategy is generic and hence also applicable to other neural operators.

[†] With a slight abuse of notation, in the following contents we use $\theta^\eta$ to denote $\theta_P^\eta$.

## 3.2 DISENTANGLING THE UNDERLYING MECHANISM

The proposed DisentangO aims to identify and disentangle the components of the latent representation $\boldsymbol{z}$, which serves as an inverse PDE solver. However, to capture the true physical mechanism, a natural question arises: is it really possible to identify the latent variables of interest (i.e., $\boldsymbol{z}$) with only observational data $\mathcal{D}_{\mathrm{tr}}$? We now show that, by learning the model $(p_{\hat{\boldsymbol{z}}}, \hat{g}, \hat{\mathcal{H}}, \hat{\mathcal{G}})$ that matches the true marginal data distribution in all domains, we can indeed achieve this identifiability for the generating process proposed in equation 3.5 and equation 3.6. Before the formal theorem, we first state our assumptions:

**Assumption 1.** (Density Smoothness and Positivity) $p_\theta$ and $p_{\boldsymbol{z}}$ are both smooth and positive.

**Assumption 2.** (Invertibility) The task-wise parameter $\theta$ can be generated by $\boldsymbol{z}$ through an invertible and smooth function $\mathcal{H}^{-1}$. Moreover, for each given $\boldsymbol{f}$, we denote $\mathcal{G}_f(\theta) := \mathcal{G}[\boldsymbol{f}; \theta](\boldsymbol{x})$ as the operator mapping from $\mathbb{R}^{d_\theta}$ to $\mathcal{U}$. $\mathcal{G}_f$ is also one-to-one with respect to $\theta$.

**Assumption 3.** (Conditional Independence) Conditioned on $\boldsymbol{b}$, each component of $\boldsymbol{z}$ is independent of each other: $\log p_{\boldsymbol{z}|\boldsymbol{b}}(\boldsymbol{z}|\boldsymbol{b}) = \sum_{i=1}^{d_z} \log p_{z_i|\boldsymbol{b}}(z_i|\boldsymbol{b})$.

**Assumption 4.** (Linear independence) There exists $2d_z + 1$ values of $\boldsymbol{b}$, such that the $2d_z$ vectors $w(\boldsymbol{z}, \boldsymbol{b}^j) - w(\boldsymbol{z}, \boldsymbol{b}^0)$ with $j = 1, \cdots, 2d_z$ are linearly independent. Here,

$$w(\boldsymbol{z}, \boldsymbol{b}^j) := \left( \frac{\partial q_1(z_1, \boldsymbol{b}^j)}{\partial z_1}, \cdots, \frac{\partial q_{d_z}(z_{d_z}, \boldsymbol{b}^j)}{\partial z_{d_z}}, \frac{\partial^2 q_1(z_1, \boldsymbol{b}^j)}{\partial z_1^2}, \cdots, \frac{\partial^2 q_{d_z}(z_{d_z}, \boldsymbol{b}^j)}{\partial z_{d_z}^2} \right), \quad (3.15)$$

with $q_i(z_i, \boldsymbol{b}) := \log p_{z_i|\boldsymbol{b}}$.

First, we show that the latent variable $\boldsymbol{z}$ can be identified up to an invertible component-wise transformation: for the true latent variable $\boldsymbol{z} \in \mathbb{R}^{d_z}$, there exists an invertible function $h : \mathbb{R}^{d_z} \to \mathbb{R}^{d_z}$, such that $\hat{\boldsymbol{z}} = h(\boldsymbol{z})$.

**Theorem 1.** We follow the data-generating process in equation 3.5 and equation 3.6, and Assumptions 1-2. Then, by learning $(p_{\hat{\boldsymbol{z}}}, \hat{g}, \hat{\mathcal{H}}, \hat{\mathcal{G}})$ to achieve:

$$p_{\hat{\boldsymbol{u}}|\boldsymbol{f}} = p_{\boldsymbol{u}|\boldsymbol{f}}, \quad (3.16)$$

where $\boldsymbol{u}$ and $\hat{\boldsymbol{u}}$ are generated from the true process and the estimated model, respectively, $\boldsymbol{z}$ is identifiable up to an invertible function $h$.

**Proof:** Please see Appendix A.

Additionally, with additional assumptions on conditional independence and datum variability, we can further obtain the following theoretical results on component-wise identifiability: for each true component $z_i$, there exists a corresponding estimated component $\hat{z}_j$ and an invertible function $h_i : \mathbb{R} \to \mathbb{R}$, such that $z_i = h_i(\hat{z}_j)$.

**Theorem 2.** We follow the data-generating process in equation 3.5 and equation 3.6 and Assumptions 1-4. Then, by learning $(p_{\hat{\boldsymbol{z}}}, \hat{g}, \hat{\mathcal{H}}, \hat{\mathcal{G}})$ to achieve equation 3.16, $\boldsymbol{z}$ is component-wise identifiable.

**Proof:** Please see Appendix A.

**Discussion on assumptions.** Intuitively, Assumptions 1-2 are required to guarantee that there exists a smooth and injective mapping between the ground-truth latent embedding $\boldsymbol{z}$ and the learned embedding $\hat{\boldsymbol{z}}$. Then, Assumption 4 indicates sufficient variability across physical systems (tasks). This is a common assumption in the nonlinear ICA literature for domain adaptation (Hyvarinen et al., 2019; Khemakhem et al., 2020; Kong et al., 2023). Further discussions and validation on these assumptions are provided in Appendix A. To our best knowledge, this is the first time the component-wise identifiability results are discussed in the context of multi-task neural operator learning.

## 3.3 A GENERIC ALGORITHM FOR DIFFERENT SUPERVISION SCENARIOS

In this section, we discuss the practical utility of DisentangO under three scenarios:

- (SC1: Supervised) The value of $\boldsymbol{b}^\eta$ is given.
- (SC2: Semi-supervised) The value of $\boldsymbol{b}^\eta$ is not given, but a label $c(\boldsymbol{b}^\eta)$ (e.g., a classification of $\boldsymbol{b}^\eta$) is given.
- (SC3: Unsupervised) No value or label is given for $\boldsymbol{b}^\eta$ for each task.

The architecture is shown in Figure 1, with a pseudo algorithm provided in Algorithm 1 of Appendix.

To obtain the posterior $q$ in equation 3.11, we assume that each latent variable satisfies a Gaussian distribution of distinct means and diagonal covariance, i.e., $q_{\mu_z, \Sigma_z}(\hat{\boldsymbol{z}}|\theta) := \mathcal{N}(\mu_z(\theta_P), \Sigma_z^2(\theta_P))$,

then estimate its mean and covariance using an MLP. As a result, the KL-divergence term in the ELBO admits a closed form:

$$\text{SC2/SC3: } D_{KL}\left(q(\hat{\boldsymbol{z}}|\theta)||p(\boldsymbol{z})\right) = \frac{1}{2}\sum_{i=1}^{d_z}\left((\Sigma_z)_i^2 + (\mu_z)_i^2 - 2\log((\Sigma_z)_i) - 1\right).$$

In the supervised setting, we take $\boldsymbol{z} = \boldsymbol{b}$, and set $\mu_z(\theta_P^\eta) = \boldsymbol{b}^\eta$, then:

$$\text{SC1: } D_{KL}\left(q(\hat{\boldsymbol{z}}|\theta)||p(\boldsymbol{z})\right) = \frac{1}{2}\sum_{i=1}^{d_z}\left((\Sigma_z)_i^2 + (\mu_z - \boldsymbol{b})_i^2 - 2\log((\Sigma_z)_i) - 1\right).$$

For the first decoder, the likelihood $p(\theta_P^\eta|\boldsymbol{z}^\eta)$ is a factorized Gaussian with mean $\mu_\theta$ and covariance $\Sigma_\theta$, computed from another MLP. By taking as input Monte Carlo samples once for each $\boldsymbol{z}$, the reconstruction accuracy term can be approximated as:

$$E_{q(\boldsymbol{z}|\theta_P)}\log p(\theta_P|\boldsymbol{z}) \approx -\sum_{i=1}^{d_\theta}\left(\frac{((\theta_P)_i - (\mu_\theta)_i)^2}{2(\Sigma_\theta)_i^2} + \log(\Sigma_\theta)_i + c\right),$$

where $d_\theta$ is the dimension of $\theta_P$, and $c$ is a constant.

For the second decoder, we parameterize it as a multi-task NO in equation 3.14, and model the discrepancy between $\hat{\mathcal{G}}_{[\hat{\theta}_P^\eta,\theta_I,\theta_Q]}[\boldsymbol{g}_j^\eta](\boldsymbol{x}_k)$ and the ground truth $\boldsymbol{u}(\boldsymbol{x}_k)$ as an additive independent unbiased Gaussian random noise $\epsilon$, with

$$\hat{\mathcal{G}}_{[\hat{\theta}_P^\eta,\theta_I,\theta_Q]}[\boldsymbol{g}_j^\eta](\boldsymbol{x}_k)) = \boldsymbol{u}_j^\eta(\boldsymbol{x}_k) + \epsilon_{\eta,j,k}, \quad \epsilon_{\eta,j,k} \sim \mathcal{N}(0,\varpi^2),$$

where $\varpi$ is a tunable hyperparameter. Then, with a uniform spatial discretization of size $\Delta x$ in a domain $\Omega \subset \mathbb{R}^{d_u}$, the log likelihood after eliminating the constant terms can be written as:

$$\frac{1}{2\varpi^2\Delta x^{2d_u}}\sum_{\eta=1}^{S}\sum_{j=1}^{n_{train}}\left\|\hat{\mathcal{G}}_{[\hat{\theta}_P^\eta,\theta_I,\theta_Q]}[\boldsymbol{g}_j^\eta](\boldsymbol{x}_k) - \boldsymbol{u}_j^\eta(\boldsymbol{x}_k)\right\|_{L^2(\Omega)}^2.$$

In our empirical tests in Section 4, for the unsupervised and semi-supervised scenarios we choose $q$ as a standard Gaussian distribution following the independence assumption of Theorem 2. Additionally, to avoid overparameterization, we take $\Sigma_\theta = \sigma_\theta^2\mathbf{I}$, with $\sigma_\theta$ being a tunable hyperparameter. In this case, the total objective can be further simplified as:

$$L_{loss} = \frac{1}{S}\sum_{\eta=1}^{S}\left(\beta_d\sum_{j=1}^{n_{train}}\left\|\hat{\mathcal{G}}_{[\hat{\theta}_P^\eta,\theta_I,\theta_Q]}[\boldsymbol{g}_j^\eta] - \boldsymbol{u}_j^\eta\right\|_{L^2(\Omega)}^2 + \left\|\hat{\theta}_P^\eta - \mu_\theta^\eta\right\|^2 + \beta_{KL}||\mu_{\boldsymbol{z}}^\eta||^2\right),$$

for the unsupervised scenario, and

$$L_{loss} = \frac{1}{S}\sum_{\eta=1}^{S}\left(\beta_d\sum_{j=1}^{n_{train}}\left\|\hat{\mathcal{G}}_{[\hat{\theta}_P^\eta,\theta_I,\theta_Q]}[\boldsymbol{g}_j^\eta] - \boldsymbol{u}_j^\eta\right\|_{L^2(\Omega)}^2 + \left\|\hat{\theta}_P^\eta - \mu_\theta^\eta\right\|^2 + \beta_{KL}||\mu_{\boldsymbol{z}}^\eta - \boldsymbol{b}^\eta||^2\right),$$

for the fully supervised scenario. Here, $\beta_d := \frac{\sigma_\theta^2}{\varpi^2\Delta x^{2d_u}}$ and $\beta_{KL} := \sigma_\theta^2$ are treated as tunable hyperparameters. In the semi-supervised scenario, we incorporate partial supervision by equipping the above loss with a constraint following Locatello et al. (2020):

$$L_{loss} = \frac{1}{S}\sum_{\eta=1}^{S}\left(\beta_d\sum_{j=1}^{n_{train}}\left\|\hat{\mathcal{G}}_{[\hat{\theta}_P^\eta,\theta_I,\theta_Q]}[\boldsymbol{g}_j^\eta] - \boldsymbol{u}_j^\eta\right\|_{L^2(\Omega)}^2 + \left\|\hat{\theta}_P^\eta - \mu_\theta^\eta\right\|^2 + \beta_{KL}||\mu_{\boldsymbol{z}}^\eta||^2 + \beta_{cls}L_c(\boldsymbol{z}^\eta, c(\beta^\eta))\right)$$

where $\beta_{cls}$ is a tunable hyperparameter, and $L_c(\boldsymbol{z}^\eta, c(\beta^\eta))$ is the corresponding loss term depending on the provided information. For instance, when taking $c$ as the classification label of $\beta^\eta$, $L_c$ can be taken as the cross-entropy loss.

We note that the coefficient $\beta_{KL}$ is closely connected to the adjustable hyperparameter in $\beta-$VAE (Higgins et al., 2017). Taking a larger $\beta_{KL}$ encourages disentanglement (Locatello et al., 2019), while it also adds additional constraint on the implicit capacity of the latent bottleneck that results in information loss (Burgess et al., 2018). On the other hand, the data reconstruction term forces the latent factors to contribute to the "reconstruction" of the more complicated solution field in a global perspective, and hence increasing $\beta_d$ is anticipated to alleviate information loss. In our empirical studies, we demonstrate the interplay between these two tunable hyperparameters.

## 4 EXPERIMENTS

We assess the performance of DisentangO across a variety of physics modeling and discovery datasets. In the following, we denote a DisentangO model with a latent dimension of $n$ as

DisentangO-$n$. Our evaluation focuses on the following key aspects. Firstly, we demonstrate the capability of DisentangO in forward PDE learning, and compare its performance with a number of relevant baselines. In particular, we select five NO-related baselines, i.e., FNO (Li et al., 2020c), NIO (Molinaro et al., 2023), MetaNO (Zhang et al., 2023), FUSE (Lingsch et al., 2024) and its extension, as well as four non-NO baselines, i.e., InVAErt (Tong et al., 2024), two VAE variants (Kingma & Welling, 2013) and $\beta$-VAE (Higgins et al., 2017). Secondly, we showcase the merits of DisentangO in inverse modeling for interpretable physics discovery. Thirdly, we perform parametric studies on the associated disentanglement hyperparameters. Lastly, we provide interpretations of the disentangled parameters and point out future directions. Further details are provided in Appendix B.

## 4.1 SUPERVISED FORWARD AND INVERSE PDE LEARNING

We start by investigating DisentangO's capability in solving both forward and inverse PDE problems in a fully supervised setting. Specifically, we consider the constitutive modeling of anisotropic fiber-reinforced hyperelastic materials governed by the Holzapfel–Gasser–Ogden (HGO) model, where the data-generating process is controlled by sampling the governing material parameter set $\{E, \nu, k_1, k_2, \alpha\}$ and the latent factors can be learned consequently in a supervised fashion. In this setting, the model takes as input the padded traction field and learns to predict the displacement.

Table 1: Test errors and the number of trainable parameters in experiment 1. Bold number highlights the best method.

| Models | #param | per-epoch time (s) | Test errors | |
|---|---|---|---|---|
| | | | data | latent $z$ (SC1) |
| DisentangO | 697,383 | 12.2 | 1.65% | **4.63%** |
| MetaNO | 295,842 | 9.8 | **1.59%** | - |
| NIO | 709,260 | 5.6 | - | 15.16% |
| FNO | 698,284 | 9.1 | 2.45% | 14.55% |
| InVAErt | 706,761 | 0.1 | - | 5.16% |
| FUSE | 706,151 | 2.2 | - | 4.99% |
| FUSE-f | 707,271 | 11.4 | 16.33% | 6.19% |
| VAE | 697,672 | 2.8 | 49.97% | 16.34% |
| convVAE | 663,551 | 2.8 | 81.11% | 16.27% |
| $\beta$-VAE | 697,672 | 2.8 | 51.16% | 16.47% |

As the number of latent factors is fixed for supervised learning in this experiment, we defer our ablation study to the second experiment. We report in Table 1 our experimental results. Note that only FNO, FUSE-f, VAE and its variations are able to handle both forward and inverse problems, whereas MetaNO can only solve the forward problem, and NIO, InVAErt, and FUSE can only solve the inverse problem. With this caveat in mind, MetaNO achieves the best performance in forward PDE learning, with DisentangO performing comparably well and beating the third best model by 32.7% in accuracy. In terms of inverse modeling with full latent supervision (SC1), DisentangO is the only method that can hold the error well below 5%, outperforming the second best method by 7.2% and the second best joint (i.e., simultaneous forward and inverse) solver by 25.2%.

## 4.2 SEMI-SUPERVISED MECHANICAL MNIST BENCHMARK

We next explore in the context of semi-supervised learning and apply DisentangO and relevant baselines to the benchmark Mechanical MNIST (MMNIST) dataset (Lejeune, 2020). MMNIST comprises 70,000 heterogeneous material specimens undergoing large deformation, each governed by a material model of the Neo-Hookean type with a varying modulus converted from the MNIST bitmap images. In our experiment, we take 500 images (with a 420/40/40 split for training/validation/test) and generate 200 loading/response data pairs per sample on a $29 \times 29$ structured grid, simulating uniaxial extension, shear, equibiaxial extension, and confined compression load scenarios. Since only partial knowledge is available for each image (i.e., the corresponding digit), we apply semi-supervised learning to the latent factors to classify the digits.

Table 2: Test errors and number of trainable parameters in experiment 2. DisentangO is abbreviated as DNO due to space limit. Bold number highlights the best method that can handle both forward and inverse settings.

| Models | | DNO-2 | DNO-5 | DNO-10 | DNO-15 | VAE | $\beta$-VAE | MetaNO |
|---|---|---|---|---|---|---|---|---|
| #param (M) | | 0.66 | 0.97 | 1.49 | 2.02 | 2.02 | 2.02 | 0.35 |
| | $\beta_d = 1$ | 12.82% | 9.56% | 7.36% | 6.29% | 16.34% | 17.13% | 2.68% |
| Error | $\beta_d = 10$ | 11.51% | 9.16% | 6.62% | 5.95% | - | - | - |
| | $\beta_d = 100$ | 11.49% | 8.43% | 6.65% | **5.48%** | - | - | - |
| | $\beta_d = 1000$ | 11.62% | 8.22% | 6.50% | 5.80% | - | - | - |

**Ablation study.** Firstly, we investigate DisentangO's predictability in forward PDE learning by comparing its performance to MetaNO (i.e., the base meta-learnt NO model without the hypernetwork structure). As seen in Table 2, MetaNO achieves a forward prediction error of 2.68%, which serves as the optimal bound for DisentangO. As we increase the latent dimension in DisentangO

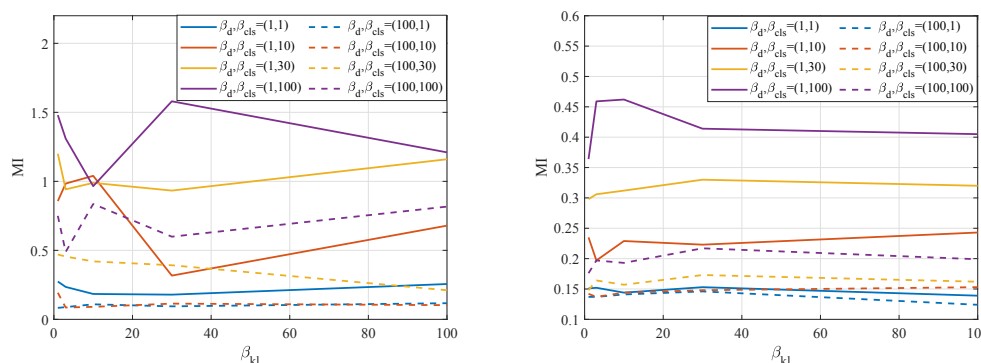

Figure 2: MMNIST unsupervised scores against $\beta_d$ with DisentangO-2 (left) and DisentangO-15 (right). By comparing $\beta_d = 1$ (solid lines) with $\beta_d = 100$ (dashed lines), increasing $\beta_d$ forces the latent factors to maximize the contained information and in turn decreases MI, thus encouraging disentanglement.

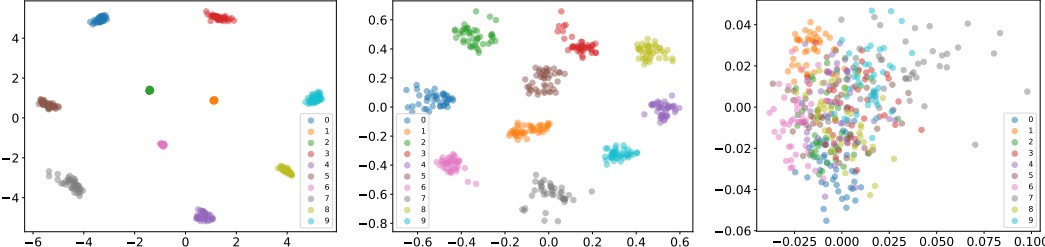

Figure 3: MMNIST scatterplot with DisentangO-2 and $\beta_d = 1$: left: $(\beta_{kl} = 1, \beta_{cls} = 100)$ (data test error 18.81%), middle: $(\beta_{kl} = 10, \beta_{cls} = 10)$ (data test error 16.94%), right: fully unsupervised DisentangO-2 without classification loss $(\beta_{kl} = 100)$ (data test error 12.65%).

from 2 to 15, the prediction error drops from 11.49% to 5.48% and converges to the optimal bound. Next, we study the role of the data loss term in disentanglement by gradually varying $\beta_d$ from $\beta_d = 1$ to $\beta_d = 1000$. In Table 2 we observe a consistent improvement in accuracy with the increase in $\beta_d$, where the boost in accuracy becomes marginal or slightly deteriorates beyond $\beta_d = 100$. We thus choose $\beta_d = 100$ as the best DisentangO model in this case. Besides offering an increased accuracy in forward modeling, the data loss term also enhances disentanglement as discussed in Section 3.2. This is evidenced by the illustration in Figure 2, where the unsupervised mutual information (MI) score that measures the amount of MI across latent factors consistently decreases in both DisentangO-2 and DisentangO-15 as we increase $\beta_d$ from $\beta_d = 1$ (solid lines) to $\beta_d = 100$ (dashed lines). On the contrary, the classification term poses a negative effect on disentanglement, as indicated by the increase in $\beta_{cls}$ leading to an increase in MI scores. This is reasonable because the classifier linearly combines all the latent factors to make a classification. The more accurate the classification, the stronger the correlation across latent factors. We then move on to study the effect of semi-supervised learning by comparing the model's performance with and without latent partial supervision. While the models without semi-supervision is slightly more accurate in that the forward prediction accuracy with $\beta_d = 1$ reaches 12.60% and 6.16% in DisentangO-2 and DisentangO-15, respectively, the latent scatterplot in Figure 3 reveals the inability of the DisentangO model without latent semi-supervision to acquire the partial knowledge of the embedded digits from data. In contrast, although the accuracy of DisentangO with latent semi-supervision slightly deteriorates due to the additional regularization effect from the classification loss term, it is able to correctly recognize the embedded digits and leverage this partial knowledge in disentangling meaningful latent factors.

**Comparison with additional baselines.** We compare the performance of DisentangO with relevant additional baselines, i.e., VAE and $\beta$-VAE. We abandon the three NO-related baselines in this and the following experiments because they are black-box approximators and do not possess any mechanism to extract meaningful information without supervision. In this context, even the least accurate DisentangO-2 model (i.e., with $\beta_d = 1$) with significantly fewer parameters outperforms the two selected baselines in Table 2 by 21.5% and 25.2%, respectively, with the best performing DisentangO-15 model beating the two baselines by 66.5% and 68.0% in accuracy, respectively. We do not proceed to compare the physics discovery capability between them as the baseline models are considerably inaccurate in forward prediction.

**Interpretable physics discovery.** To showcase the interpretability of DisentangO, we visualize its latent variables through latent traversal based on a randomly picked loading field as input in DisentangO-2, as displayed in Figure 6 of Appendix B.3. One can clearly see that the digit changes from "6" to "0" and then "2" from the top left moving down, and from "6" to "1" and then "7" moving to the right. Other digits are visible as well such as "7", "9", "4" and "8" in the right-most column. This corresponds well to the distribution of the latent clustering in Figure 3. More discussions on the interpretability of DisentangO-15 can be found in Figures 7 and 8 in Appendix B.3.

### 4.3 Unsupervised heterogeneous material learning

We now demonstrate the applicability of DisentangO in unsupervised learning in the context of learning synthetic human tissues that exhibit highly organized structures, with collagen fiber arrangements varying spatially. In this subject, it is critical to understand the underlying low-dimensional disentangled properties in the hidden latent space of this complex, high-dimensional microstructure, as inferred from experimental mechanical measurements. We generate two datasets, each containing 500 specimens and 100 loading/displacement pairs. The first dataset features variations in fiber orientation distributions using a Gaussian Random Field (GRF) and the second differs fiber angles in two segmented regions, separated by a centerline with a randomly rotated orientation.

Table 3: Test errors and number of trainable parameters in experiment 3. DisentangO is abbreviated as DNO due to space limit. Bold number highlights the best method that can handle both forward and inverse settings.

| Models | #param (M) | Test error | |
| --- | --- | --- | --- |
| | | $\beta_d = 1$ | $\beta_d = 100$ |
| DNO-2 | 0.63 | 26.33% | 25.18% |
| DNO-5 | 0.94 | 17.51% | 15.75% |
| DNO-10 | 1.46 | 10.76% | 10.01% |
| DNO-15 | 1.98 | 7.11% | 7.02% |
| DNO-30 | 3.55 | 5.33% | **5.28%** |
| VAE | 3.55 | 61.10% | - |
| $\beta$-VAE | 3.55 | 57.04% | - |
| MetaNO | 0.32 | 2.67% | - |

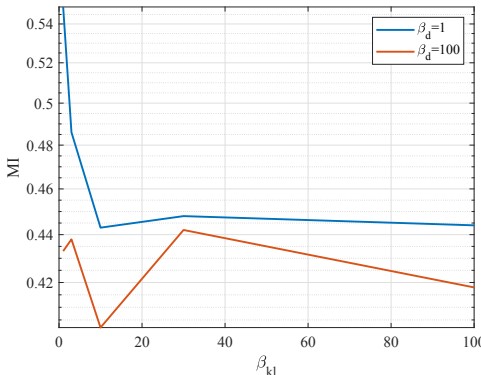

Figure 4: Unsupervised MI score against $\beta_{kl}$ with DisentangO-2 in heterogeneous material learning.

We report in Table 3 our experimental results of DisentangO with different latent dimensions and data loss strength $\beta_d$, along with comparisons with the baseline model results. Consistent with the findings in the second experiment, increasing $\beta_d$ results in a boost in accuracy by comparing the rows in the table. The model's predictive performance also converges to the optimal bound of MetaNO as we increase the latent dimension from 2 to 30, where the model's prediction error improves from 25.18% to 5.28%. This significantly outperforms the best baseline model by 90.7%. On the other hand, the effect of the data loss term on disentanglement is further proved in Figure 4, where increasing $\beta_d$ leads to a decrease in MI between the latent factors, thus encouraging disentanglement. Lastly, we interpret the mechanism of the learned latent factors in DisentangO-3 via learning a mapping between the learned latent factors and the underlying material microstructure and subsequently performing latent traversal in each dimension. The results are shown in Figure 5, where the three latent factors manifest control on the border rotation between the two segments, the relative fiber orientation between the two segments, and the fiber orientation of the top segment, respectively.

## 5 Conclusion

We present DisentangO, a novel hyper-neural operator architecture designed to disentangle latent physical factors embedded within black-box neural operator parameters. DisentangO leverages a hypernetwork-type neural operator architecture that extracts varying parameters of the governing PDE through a task-wise adaptive layer, then further disentangles these variations into distinct latent factors. By learning these disentangled representations, DisentangO not only enhances physical interpretability but also enables more robust generalization across diverse physical systems. We perform empirical evaluations across supervised, semi-supervised, and unsupervised learning contexts.

**Limitations:** Due to limited computational resource, our experiments are focused on learning a small-to-medium number ($< 500$). Expanding the coverage to encompass a broader range of physical systems and enabling learning across more diverse parameters would be beneficial.

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

# A  IDENTIFIABILITY ANALYSIS

## A.1  PROOF OF THE MAIN THEOREMS

We first provide the proof of Theorem 1:

**Proof:** With equation 3.16, we have:

$$p_{\mathcal{G}[\boldsymbol{f};\theta]|\boldsymbol{f}} = p_{\hat{\mathcal{G}}[\boldsymbol{f};\hat{\theta}]|\boldsymbol{f}} \Leftrightarrow p_{\mathcal{G}_f(\theta)|\boldsymbol{f}} = p_{\hat{\mathcal{G}}_f(\hat{\theta})|\boldsymbol{f}} \Leftrightarrow p_{\theta|\boldsymbol{f}} = p_{\mathcal{G}_f^{-1}\circ\hat{\mathcal{G}}_f(\hat{\theta})|\boldsymbol{f}}.$$

Note that the parameter $\theta$ varies with the change of $\boldsymbol{b}$. Per the data generating process in equation 3.6, the distribution of $\boldsymbol{b}$ is invariant to $\boldsymbol{f}$. Therefore, the distribution of $\theta$ is also invariant to $\boldsymbol{f}$:

$$p_\theta = p_{\theta|\boldsymbol{f}} = p_{\mathcal{G}_f^{-1}\circ\hat{\mathcal{G}}_f(\hat{\theta})|\boldsymbol{f}} = p_{\mathcal{G}_f^{-1}\circ\hat{\mathcal{G}}_f(\hat{\theta})}, \ \forall \boldsymbol{f} \in \mathcal{F}.$$

Denoting $r := \mathcal{G}_f \circ \hat{\mathcal{G}}_f^{-1}$, it is the transformation between the true $\theta$ and the estimated one, and it is invertible and invariant with respect to $\boldsymbol{f}$.

We proceed to derive the relation between $\boldsymbol{z}$ and $\hat{\boldsymbol{z}}$: since $\theta = r(\hat{\theta})$, with the invertibility assumption $\theta = \mathcal{H}^{-1}(\boldsymbol{z})$ and $\hat{\theta} = \hat{\mathcal{H}}^{-1}(\hat{\boldsymbol{z}})$, we obtain:

$$\boldsymbol{z} = \mathcal{H}(\theta) = \mathcal{H} \circ r(\hat{\theta}) = \mathcal{H} \circ r \circ \hat{\mathcal{H}}^{-1}(\hat{\boldsymbol{z}}).$$

Denoting $h := \mathcal{H} \circ r \circ \hat{\mathcal{H}}^{-1}$, it is the transformation between the true latent variable and the estimated one, and it is invertible because $r$, $\mathcal{H}$ and $\hat{\mathcal{H}}$ are all invertible. $\qquad\square$

We now show the proof of Theorem 2.

**Proof:** With the independence relation assumption, we have

$$p_{\boldsymbol{z}|\boldsymbol{b}}(\boldsymbol{z}|\boldsymbol{b}) = \prod_i p_{z_i|\boldsymbol{b}}(z_i), \ p_{\hat{\boldsymbol{z}}|\boldsymbol{b}}(\hat{\boldsymbol{z}}|\boldsymbol{b}) = \prod_i p_{\hat{z}_i|\boldsymbol{b}}(\hat{z}_i) \ .$$

Denoting $\hat{q}_i := \log p_{\hat{z}_i|\boldsymbol{b}}$, it yields:

$$\log p_{\boldsymbol{z}|\boldsymbol{b}}(\boldsymbol{z}|\boldsymbol{b}) = \sum_i q_i(z_i, \boldsymbol{b}), \ \log p_{\hat{\boldsymbol{z}}|\boldsymbol{b}}(\hat{\boldsymbol{z}}|\boldsymbol{b}) = \sum_i \hat{q}_i(\hat{z}_i, \boldsymbol{b}) \ .$$

With the change of variables we have

$$p_{\boldsymbol{z}|\boldsymbol{b}} = p_{h(\hat{\boldsymbol{z}})|\boldsymbol{b}} = p_{\hat{\boldsymbol{z}}|\boldsymbol{b}} \cdot |J_{h^{-1}}| \Leftrightarrow \sum_i q_i(z_i, \boldsymbol{b}) + \log |J_h| = \sum_i \hat{q}_i(\hat{z}_i, \boldsymbol{b}) \ ,$$

where $|J_{h^{-1}}|$ stands for the absolute value of the Jacobian matrix determinant of $h^{-1}$. Differentiating the above equation twice with respect to $\hat{z}_k$ and $\hat{z}_q$, $k \neq q$, yields

$$\sum_i \left( \frac{\partial^2 q_i(z_i, \boldsymbol{b})}{\partial z_i^2} \frac{\partial z_i}{\partial \hat{z}_k} \frac{\partial z_i}{\partial \hat{z}_q} + \frac{\partial q_i(z_i, \boldsymbol{b})}{\partial z_i} \frac{\partial^2 z_i}{\partial \hat{z}_k \partial \hat{z}_q} \right) + \frac{\partial^2 \log |J_h|}{\partial \hat{z}_k \partial \hat{z}_q} = 0 \ . \qquad (\text{A.1})$$

To show the identifiability, one can rewrite the Jacobian $J_h$ as:

$$J_h = \left[ \frac{\partial \boldsymbol{z}}{\partial \hat{\boldsymbol{z}}} \right] \ .$$

The invertibility results shown in Theorem 1 indicates that it is full rank. Next, we will use the linear independence assumption to show that there exists one and only one non-zero component in each row of $\frac{\partial \boldsymbol{z}}{\partial \hat{\boldsymbol{z}}}$.

Taking $\boldsymbol{b} = \boldsymbol{b}^0, \cdots, \boldsymbol{b}^{2d_z}$ in equation A.1 and subtracting them from each other, we have

$$\sum_{i=1}^{d_z} \left( \left( \frac{\partial^2 q_i(z_i, \boldsymbol{b}^j)}{\partial z_i^2} - \frac{\partial^2 q_i(z_i, \boldsymbol{b}^0)}{\partial z_i^2} \right) \frac{\partial z_i}{\partial \hat{z}_k} \frac{\partial z_i}{\partial \hat{z}_q} + \left( \frac{\partial q_i(z_i, \boldsymbol{b}^j)}{\partial z_i} - \frac{\partial q_i(z_i, \boldsymbol{b}^0)}{\partial z_i} \right) \frac{\partial^2 z_i}{\partial \hat{z}_k \partial \hat{z}_q} \right) = 0 \ ,$$

where $j = 1, \cdots, 2d_z$. With the linear independence condition for $w$, this is a $2d_z \times 2d_z$ linear system, and therefore the only solution is

$$\frac{\partial z_i}{\partial \hat{z}_k} \frac{\partial z_i}{\partial \hat{z}_q} = 0 \,, \quad \frac{\partial^2 z_i}{\partial \hat{z}_k \partial \hat{z}_q} = 0 \,,$$

for $i = 1, \cdots, d_z$. The first part implies that, for the $i-$th row of the Jacobian matrix $J_h$, we have $\dfrac{\partial z_i}{\partial \hat{z}_k} \neq 0$ for at most one element $k \in \{1, \cdots, d_z\}$, hence $\boldsymbol{z}$ is identifiable up to permutation and component-wise invertible transformation. $\qquad \square$

## A.2 Further discussion on the assumptions

Herein, we provide additional discussion on the validity and emprical validation for Assumptions 1-4.

As seen in the proof above, Assumptions 1 (Density Smoothness and Positivity) and 2 (Invertibility) are required to guarantee that there exists a smooth and injective mapping $h := \mathcal{H} \circ r \circ \hat{\mathcal{H}}^{-1}$, from the ground-truth latent embedding $\boldsymbol{z}$ to the learned embedding $\hat{\boldsymbol{z}}$. Furthermore, the smoothness assumption further makes it feasible to take derivatives of $\boldsymbol{z}$ with respect to $\hat{\boldsymbol{z}}$, which supports the permutation-wise identifiability proof for Theorem 2. Here, we note that the smoothness assumption may possibly be relaxed to $C^2$. Assumptions 3 (Conditional Independence) and 4 (Linear Independence) are needed to show that the Jacobian of $h$ has one and only one non-zero component for each column. Without these assumptions, it is possible that the data from different $\mathbf{b}$ lack variability.

While the first three assumptions are common in many VAE architectures, the last assumption is plausible for many real-world data distributions. For instance, when the prior on the latent variables $p(\boldsymbol{z}|\boldsymbol{b})$ is conditionally factorial, where each element $z_i$ has a univariate exponential family distribution given conditioning variable $\boldsymbol{b}$:

$$p(\boldsymbol{z}|\boldsymbol{b}) = \prod_i \frac{Q_i(z_i)}{Z_i(\boldsymbol{b})} \exp\left[\sum_{j=1}^{k} T_{i,j}(z_i)\lambda_{i,j}(\boldsymbol{b})\right],$$

where $Q_i$ is the base measure, $Z_i(\boldsymbol{b})$ is the normalizing constant, $T_{i,j}$ are the sufficient statistics, and $\lambda_{i,j}$ are the corresponding parameters depending on $\boldsymbol{b}$. This exponential family has universal approximation capabilities. Additionally, we note that this distribution is conditionally independent with

$$q_i = \log(Q_i(z_i)) - \log(Z_i(\boldsymbol{b})) + \left[\sum_{j=1}^{k} T_{i,j}(z_i)\lambda_{i,j}(\boldsymbol{b})\right],$$

and the linear independence indicates that the matrix formed by

$$\omega(\boldsymbol{z}, \boldsymbol{b}^\eta) - \omega(\boldsymbol{z}, \boldsymbol{b}^0) = \left(\mathbf{T}'(\lambda(\boldsymbol{b}^\eta) - \lambda(\boldsymbol{b}^0)), \mathbf{T}''(\lambda(\boldsymbol{b}^\eta) - \lambda(\boldsymbol{b}^0))\right), \eta = 1, \cdots, S$$

has full rank $2d_z$.

In the fully supervised case, the conditional independence and linear independence assumptions are automatically guaranteed by picking proper distributions $p(\boldsymbol{z}|\boldsymbol{b})$ when designing the VAE architecture. In the semi-supervised and unsupervised cases, one can also validate these assumptions when the true values of $\boldsymbol{b}$ is given on some tasks, by inferring an empirical distribution of $p(\boldsymbol{z}|\boldsymbol{b})$ from these tasks. To investigate such a capability, in the additional synthetic experiment in Appendix C.4, we consider an unsupervised setting, estimate $p(\boldsymbol{z}|\boldsymbol{b})$ from the trained model, and check the linear independence condition by calculating the vector in equation 3.15 for each task $\boldsymbol{b}^\eta$ and forming an $S \times 2d_z$ matrix from all tasks. When the rank of this matrix is $2d_z$, it means that we can select $2d_z + 1$ tasks from them with sufficient variability, such that the linear independence condition is satisfied. As a demonstration, in Appendix C.4 we validate this assumption on a synthetic dataset and show that the identifiability can be largely achieved with Gaussian distributions of distinct means and variances.

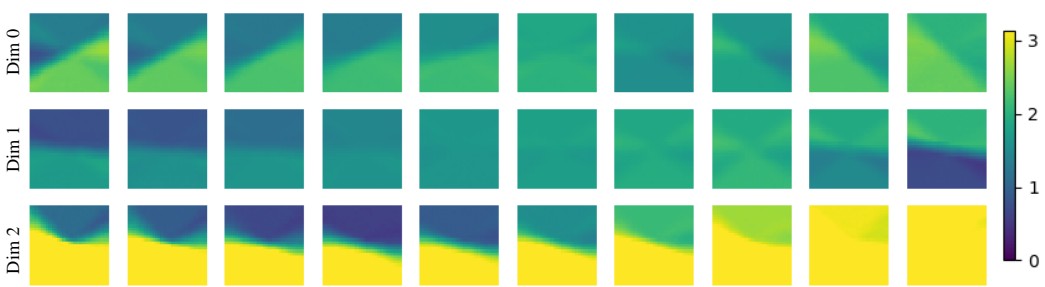

Figure 5: Latent traversal of DisentangO-3 in unsupervised heterogeneous material learning, where the three latent dimensions control the border rotation between the two segments (top), the relative fiber orientation between the two segments (middle), and the fiber orientation of the top segment (bottom), respectively. Legend indicates fiber orientation ranging from 0 to $\pi$.

## B  ADDITIONAL EXPERIMENTAL DETAILS, RESULTS AND DISCUSSION

Herein, we provide additional details in training and baseline models, as well as more results as a supplement of Section 4.

### B.1  TRAINING DETAILS

In all experiments, we adopt the Adam optimizer for optimization and use a 16-layer DisentangO model. For fair comparison across different models, we tune the hyperparameters, including the learning rates, the decay rates, and the regularization parameters, to minimize the validation loss. Experiments are conducted on a single NVIDIA Tesla A100 GPU with 40 GB memory. A pseudo algorithm of all three scenarios is summarized in Algorithm 1.

---

**Algorithm 1** A pseudo algorithm of DisentangO.

---

1: Denote data reconstruction loss $L_{data}$, task-wise NO parameter reconstruction loss $L_{recon}$, KL loss $L_{KL}$ and semi-supervision loss $L_{semi}$ as:

$$L_{data} = \frac{1}{S}\sum_{\eta=1}^{S}\left[-E(\log(p(\boldsymbol{u}|\boldsymbol{f},\theta^{\eta})))\right] , \quad L_{recon} = \frac{1}{S}\sum_{\eta=1}^{S}\left[-E_{q(\boldsymbol{z}^{\eta}|\theta^{\eta})}\log p(\theta^{\eta}|\boldsymbol{z}^{\eta})\right] ,$$

$$L_{KL} = \frac{1}{S}\sum_{\eta=1}^{S}\left[D_{KL}\left(q(\boldsymbol{z}^{\eta}|\theta^{\eta})||p(\boldsymbol{z}^{\eta})\right)\right] , \quad L_{semi} = L_c(\boldsymbol{z}^{\eta}, c(\boldsymbol{b}^{\eta})) .$$

2: **SC1: Supervised/SC3: Unsupervised**
3: The total loss is comprised of the data loss, the NO parameter reconstruction loss, and the KL loss: $L_{loss} = \beta_d L_{data} + L_{recon} + \beta_{KL}L_{KL}$.
4: **SC2: Semi-supervised**
5: The total loss is comprised of the data loss, the NO parameter reconstruction loss, the KL loss, and a semi-supervised loss: $L_{loss} = \beta_d L_{data} + L_{recon} + \beta_{KL}L_{KL} + \beta_{cls}L_{semi}$.

---

### B.2  SUPERVISED FORWARD AND INVERSE PDE LEARNING

The parameter of each baseline is given in the following, where the parameter choice of each model is selected by tuning the number of layers and the width (channel dimension), keeping the total number of parameters on the same magnitude.

- MetaNO: We use a 8-layer IFNO model with the lifting layer as the adaptive layer. We keep the total number of parameters in MetaNO the same as the number of parameters used in forward PDE learning in DisentangO.
- NIO: We closely follow the setup in Molinaro et al. (2023), where two neural operators (DeepONet and FNO) are stacked together to realize the operator-to-function intuition. The first operator maps multiple solution functions to a set of representations (which can be seen as an analog of eigenfunctions), and the second operator infers the underlying parameter

field from the mixed representations. As NIO requires the solution field as input, it cannot be used as a forward solver. Hence, NIO only solves the inverse PDE problem, and it can only be applied to the fully supervised setting. Specifically, we use four convolution blocks as the encoder for the branch net and a fully connected neural network with two hidden layers of 256 neurons as the trunk net, with the number of basis functions set to 50. For the FNO part, we use one Fourier layer with width 32 and modes 8, as suggested in Molinaro et al. (2023).

- FNO: Since FNO is originally designed as a function-to-function mapping, we consider the inverse optimization procedure following Lee et al. (2024), and develop a two-phase process to solve the forward and inverse problems sequentially. In the first phase, we construct the forward mapping from the loading field $\boldsymbol{f}$ and the ground-truth material parameter $\boldsymbol{b}$ to the corresponding solution $\boldsymbol{u}$ as: $\mathcal{G}^{FNO}[\boldsymbol{f}, \boldsymbol{b}; \theta^{FNO}](\boldsymbol{x}) = \boldsymbol{u}(\boldsymbol{x})$. This can be seen as an analog of the operator $\mathcal{G}$ in our equation 3.7. Then, with the trained FNO as a surrogate for the forward solution operator, we fix its NN parameters $\theta^{FNO}$, and use it together with gradient-based optimization to solve for the optimal material parameters as an inverse solver. Specifically, given a set of loading/solution data pairs $\{(f_i, u_i)\}_{i=1}^{N}$, we start from a random guess of the underlying material parameters (typically chosen as the average of all available instances of material parameters for fast convergence), and minimize the difference between the predicted displacement field from FNO and the ground-truth one:

$$b^* = \operatorname{argmin}_b \sum_{i=1}^{N} \|u_i - \mathcal{G}^{FNO}[f_i, b; \theta^{FNO}]\|^2.$$

As the FNO parameters are fixed, we can back propagate this loss and optimize the input material parameters in an iterative fashion. We adopt a 4-layer FNO with width 26 and modes 8. For the forward model, in addition to the loading field and the coordinates as input, we also concatenate the ground-truth material properties to form the final input. For the inverse model, we employ an iterative gradient-based optimization to solve for the optimal material parameters for physics discovery. For fair comparison, in terms of the averaged per-epoch runtime, we report the sum of both the forward and inverse solvers. The averaged per-epoch runtimes for the forward solver and the inverse solver are 6.2 seconds and 2.9 seconds, respectively, accounting for the total per-epoch runtime of 9.1 seconds in Table 1.

- FUSE: For the forward model, we take three Fourier layers in addition to the first band-limited lifting layer that increases the dimension of the parameters and performs an inverse Fourier transform. On the other hand, the inverse model maps the functional input to the parameter space by employing a concatenation of two Fourier layers and a band-limited forward Fourier transform that generates a fixed-size latent representation of the input function, followed by a flow-matching posterior estimation (FMPE) that maps to the final parameter output. All Fourier layers have a latent width of 32 and 8 modes, while the FMPE flow has 4 layers of width 360.

- FUSE-f: Since the original FUSE model assumes a constant loading field, it cannot handle situations where the input loading field changes that create multiple instances of a PDE system. We therefore create a FUSE variation (denoted as "FUSE-f") that takes a concatenation of both the displacement field and the loading field as input to the inverse model. For the forward model, we concatenate the loading field to the output of the first band-limited lifting layer and subsequently use a one-layer MLP to map it back to the original dimension. All other settings are the same as the original FUSE baseline.

- InVAErt: We directly take the InVAErt implementation from Lingsch et al. (2024) and define the encoder, the VAE encoder and the decoder as 4-layer MLPs with hidden dimension 96 and silu activation function.

- VAE: We use a 2-layer MLP of size [1681, 136, 30] as the encoder, and another 2-layer MLP of size [30, 136, 1681] as the decoder, with the size of the bottleneck layer being 30.

- convVAE: We use a convolutional layer with 136 kernels of size $3 \times 3$ with a stride of 2 pixels and a fully connected layer of size [59976, 5] as the encoder, and a fully connected layer of size of [5, 59976] and a transposed convolutional layer with 2 kernels of size $3 \times 3$ as the decoder.

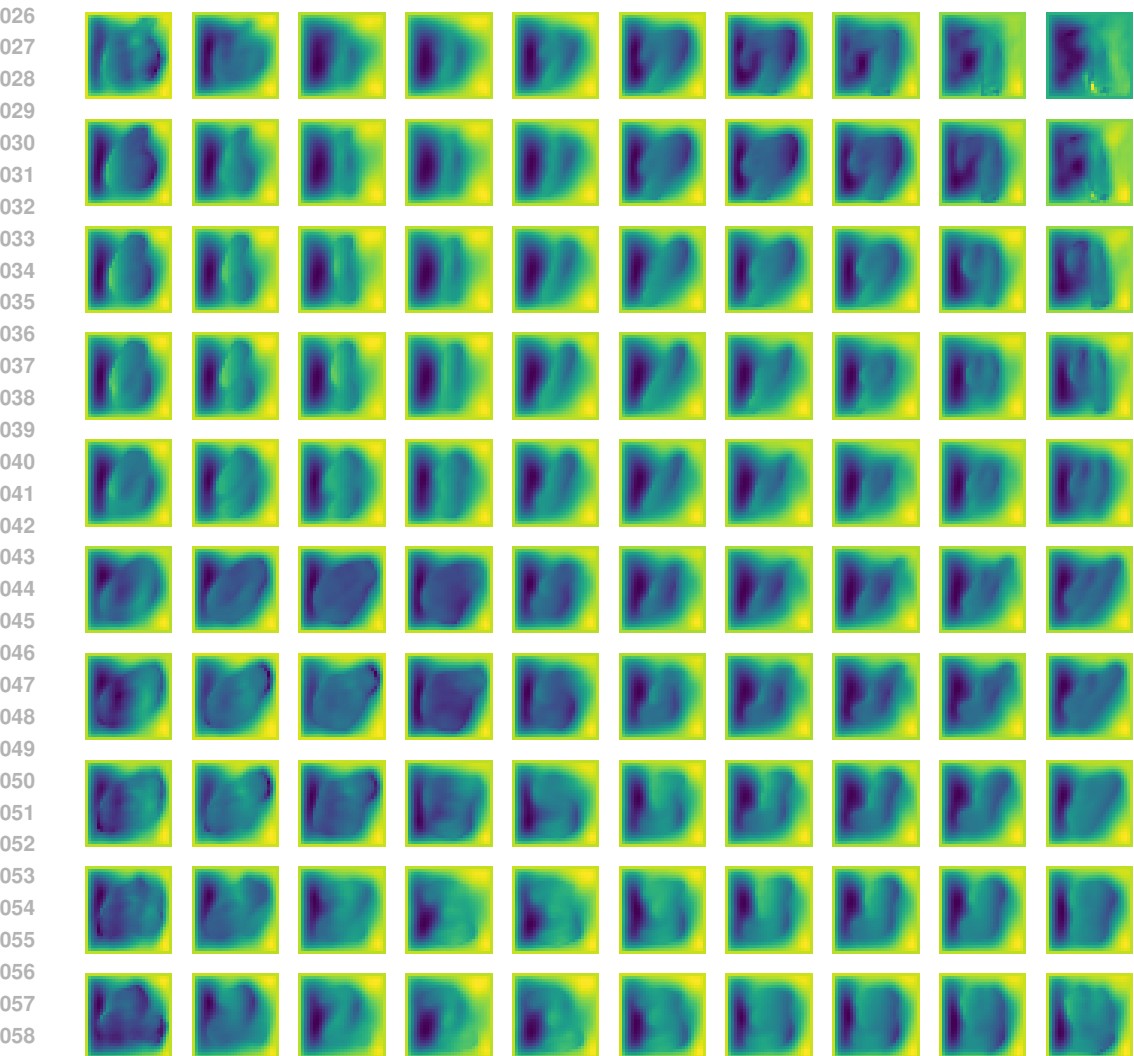

Figure 6: MMNIST latent traversal based on a randomly picked loading field as DisentangO-2 input.

- $\beta$-VAE: The parameter choice of the $\beta$-VAE baseline is the same as the VAE baseline, except that we tune the $\beta$ hyperparameter.

### B.3 MECHANICAL MNIST BENCHMARK

Herein, we demonstrate the latent traversal based on a randomly picked loading field as DisentangO-2 input. One can clearly see that the digit changes from "6" to "0" and then "2" from the top left moving down, and from "6" to "1" and then "7" moving to the right. Other digits are visible as well such as "7", "9", "4" and "8" in the right-most column. This corresponds well to the distribution of the latent clustering in Figure 3. We also provide two exemplary MMNIST latent interpretations of the learned DisentangO-15 in Figures 7 and 8.

### B.4 LATENT VISUALIZATION AND PHYSICAL INTERPRETATION

As the hidden parameter field $b^\eta$ of the PDE is generally not accessible, especially in the semi-supervised or unsupervised settings, one cannot directly reconstruct $b^\eta$ from the learned latent variables $z$. Through disentangled representation learning, the latent variables $z$ are anticipated to contain critical information of $b^\eta$, but there does not exist a direct and explicit mapping from $z$ to $b^\eta$. As a result, one cannot directly reconstruct $b^\eta$ from $z$. However, there are several tricks one can play

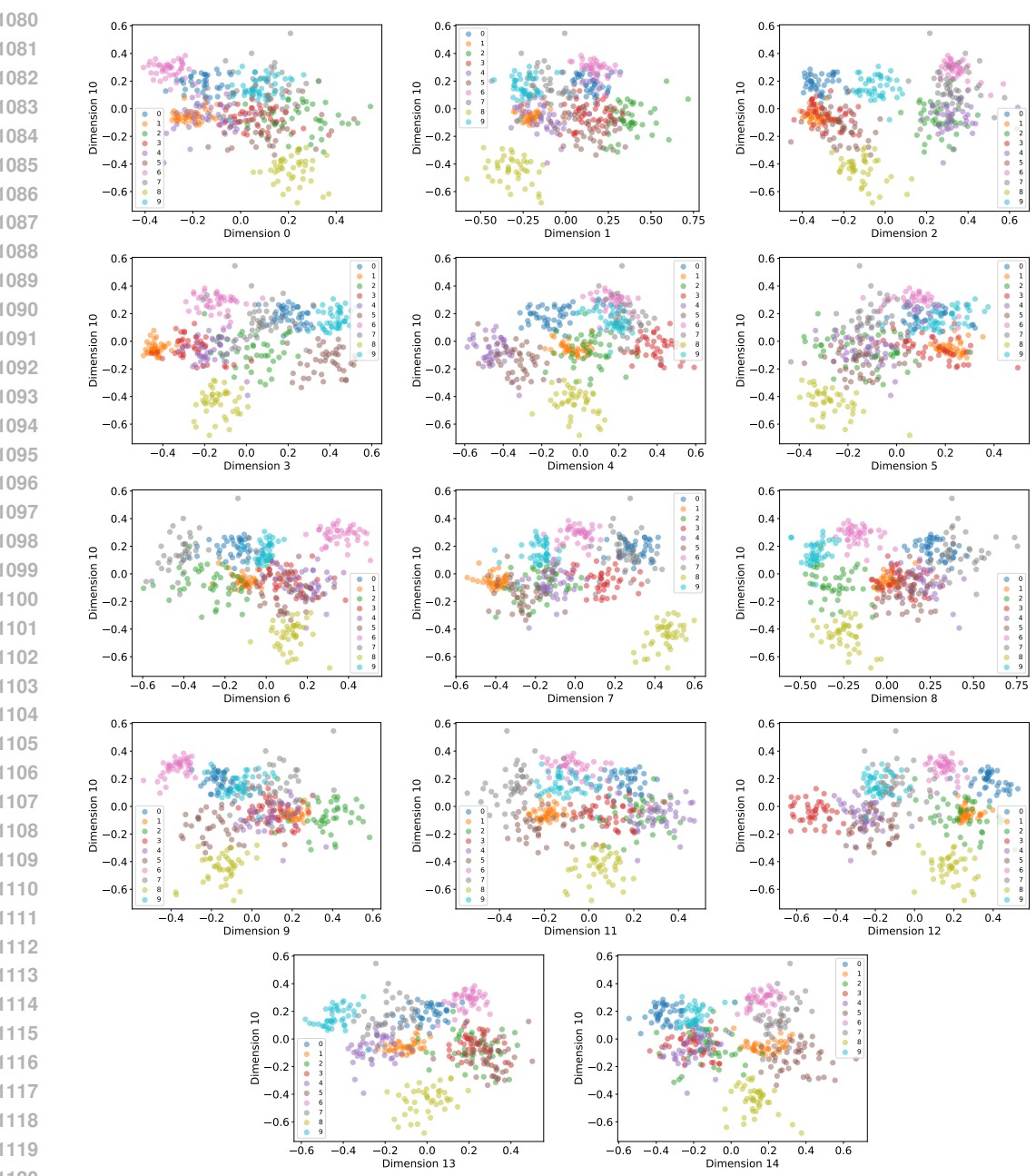

Figure 7: Exemplary MMNIST latent interpretation of DisentangO-15: dimension 10 encodes the information for digit '8', as is evidenced by the fact that all $y < -0.4$ regions on the scatterplots contain only digit '8'.

with to obtain meaningful interpretations from the learned $z$. On one hand, one can feed a randomly selected input into DisentangO and manually define a desired $z$ in the latent space and perform a forward pass using the trained DisentangO. $b^\eta$ can then be visualized from the output, as demonstrated in Figure 6 in the MMNIST experiment. On the other hand, one can also train a simple MLP and construct a mapping from $z$ to $b^\eta$, provided that $b^\eta$ is available. With this mapping at hand, one can then traverse $z$ and visualize what each dimension of $z$ controls. This is demonstrated in Figure 5 in the third experiment.

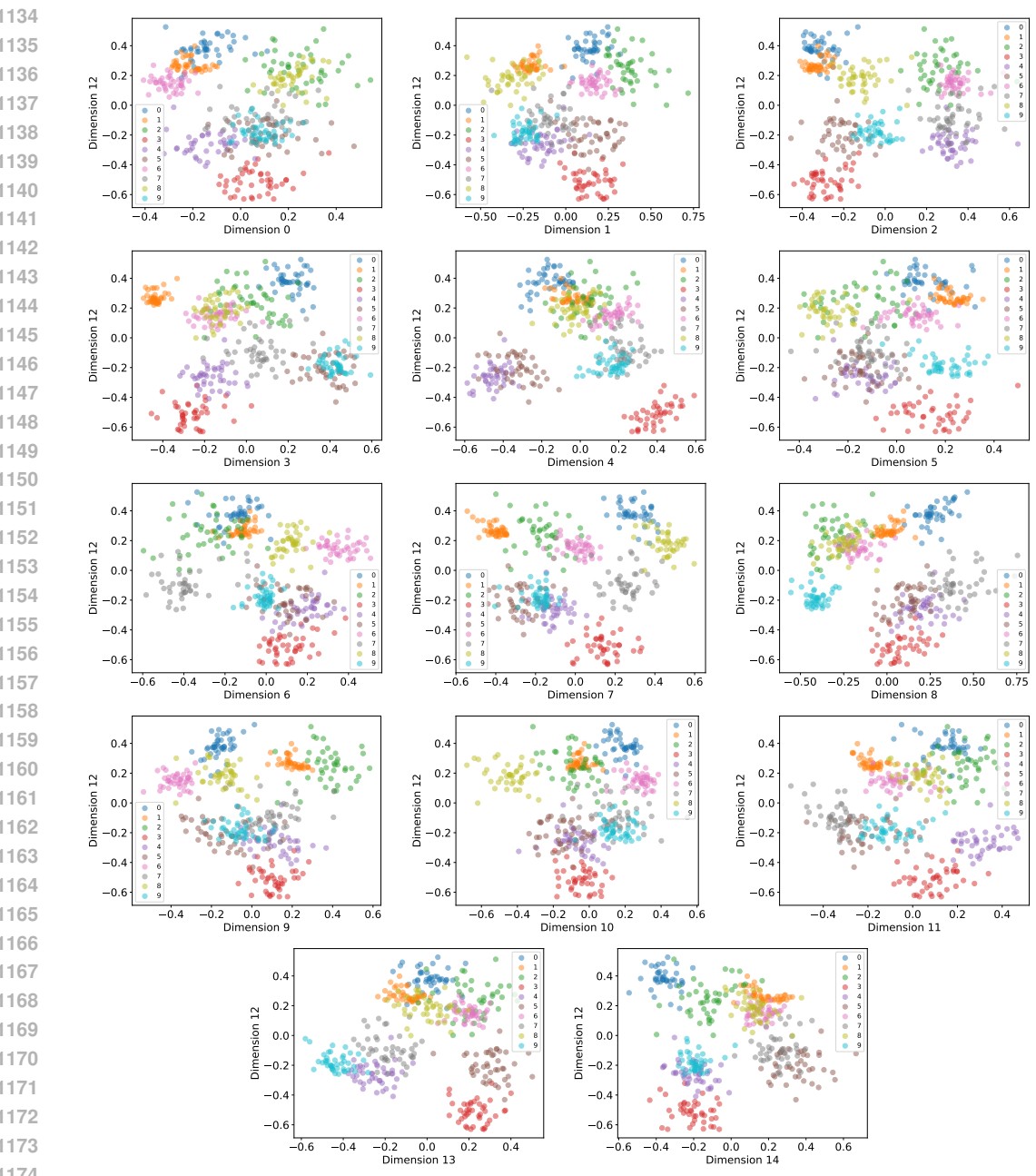

Figure 8: Exemplary MMNIST latent interpretation of DisentangO-15: dimension 12 encodes the information for digit '3', as is evidenced by the fact that all $y < -0.45$ regions on the scatterplots contain only digit '3'.

# C   DATA GENERATION

## C.1   EXPERIMENT 1 - SUPERVISED FORWARD AND INVERSE PDE LEARNING

we consider the constitutive modeling of anisotropic fiber-reinforced hyperelastic materials governed by the Holzapfel–Gasser–Ogden (HGO) model, whose strain energy density function can be

written as:

$$\eta = \frac{E}{4(1+\nu)}(\bar{I}_1 - 2) - \frac{E}{2(1+\nu)}\ln J$$

$$+ \frac{k_1}{2k_2}(e^{k_2\langle S(\alpha)\rangle^2} + e^{k_2\langle S(-\alpha)\rangle^2} - 2) + \frac{E}{6(1-2\nu)}(\frac{J^2-1}{2} - \ln J), \tag{C.1}$$

where $\langle \cdot \rangle$ indicates the Macaulay bracket, $\alpha$, $k_1$ and $k_2$ are the fiber angle, modulus and exponential coefficient, respectively, $E$ denotes the Young's modulus of the matrix, $\nu$ is Poisson's ratio, and $S(\alpha)$ describes the fiber strain of the two fiber groups, $S(\alpha) = \frac{\bar{I}_4(\alpha)-1+|\bar{I}_4(\alpha)-1|}{2}$. $\bar{I}_i$ is the $i^{\text{th}}$ invariant of the right Cauchy-Green tensor $\boldsymbol{C}$, $\bar{I}_1 = tr(\boldsymbol{C})$ and $\bar{I}_4 = \boldsymbol{n}^T(\alpha)\boldsymbol{C}\boldsymbol{n}(\alpha)$, with $\boldsymbol{n}(\alpha) = [\cos(\alpha), \sin(\alpha)]^T$. In this context, the data generation process is controlled by sampling the material set $\{E, \nu, k_1, k_2, \alpha\}$, and the latent factors can be learned consequently in a supervised fashion. The physical parameters are sampled from $\frac{k_1}{k_2} \sim \mathcal{U}[90, 100]$ ,$k_2 \sim \mathcal{U}[0.001, 0.1]$, $E \sim \mathcal{U}[0.5001, 0.6001]$, $\nu \sim \mathcal{U}[0.2, 0.3]$, and $\alpha \sim \mathcal{U}[\pi/10, \pi/2]$. To generate the high-fidelity (ground-truth) dataset, we sample 220 material sets, which are split into 200/10/10 for training/validation/test, respectively. For each material set, we sample 50 different vertical traction conditions $T_y(\boldsymbol{x})$ on the top edge from a random field, following the algorithm in Lang & Potthoff (2011b); Yin et al. (2022). $T_y(\boldsymbol{x})$ is taken as the restriction of a 2D random field, $\phi(\boldsymbol{x}) = \mathcal{F}^{-1}(\gamma^{1/2}\mathcal{F}(\Gamma))(\boldsymbol{x})$, on the top edge. Here, $\Gamma(\boldsymbol{x})$ is a Gaussian white noise random field on $\mathbb{R}^2$, $\gamma = (w_1^2 + w_2^2)^{-\frac{2}{4}}$ represents a correlation function, and $w_1, w_2$ are the wave numbers on $x$ and $y$ directions, respectively. Then, for each sampled traction loading, we solve the displacement field on the entire domain by minimizing the potential energy using the finite element method implemented in FEniCS (Alnæs et al., 2015). Sample data of the obtained dataset is illustrated in Figure 9. In this setting, the model takes as input the padded traction field and learns to predict the resulting displacement field.

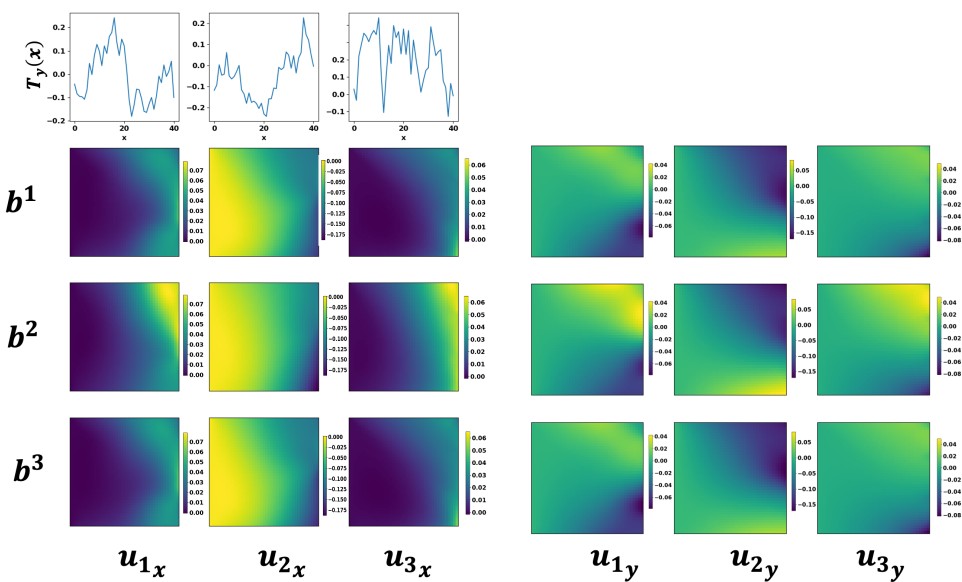

Figure 9: Illustration of the HGO data, loading and displacement pairs of three materials $\boldsymbol{b}^\eta$, $\eta = 1, 2, 3$. Top: three instances of different loadings $T_y(x)$, which corresponds to different $\boldsymbol{f}_i$. Bottom: corresponding displacement solutions $\boldsymbol{u}_i^\eta$, illustrating the impacts of system ($\boldsymbol{b}^\eta$) variability in solution operators.

## C.2 EXPERIMENT 2 - MECHANICAL MNIST BENCHMARK

Mechanical MNIST is a benchmark dataset of heterogeneous material undergoing large deformation, modeled by the Neo-Hookean material with a varying modulus converted from the MNIST bitmap images (Lejeune, 2020). It contains 70,000 heterogeneous material specimens, and each specimen is governed by the Neo-Hookean material with a varying modulus converted from the MNIST bitmap

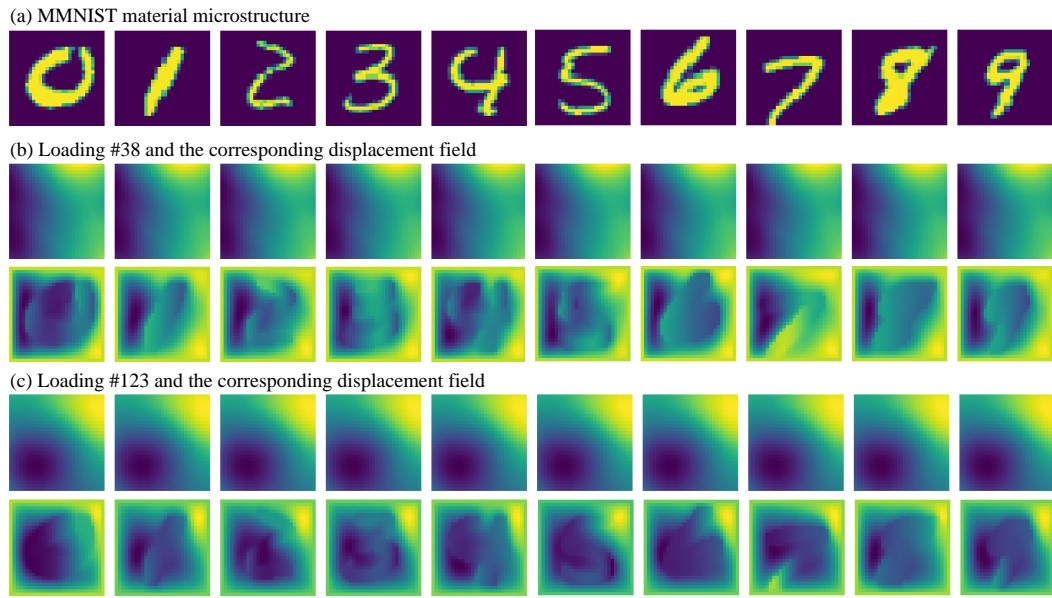

Figure 10: Illustration of exemplar MMNIST samples in the semi-supervised scenario. (a): material parameter field corresponding to different $\boldsymbol{b}^\eta$. (b): displacement fields (second row) $\boldsymbol{u}_{38}^\eta$ corresponding to the same loading field (first row) $\boldsymbol{f}_{38}$. (c): displacement fields (second row) $\boldsymbol{u}_{138}^\eta$ corresponding to the same loading field (first row) $\boldsymbol{f}_{138}$.

images. We illustrate samples from the MMNIST dataset in Figure 10, including the underlying microstructure, two randomly picked loading fields, and the corresponding displacement fields.

## C.3   EXPERIMENT 3 - UNSUPERVISED HETEROGENEOUS MATERIAL LEARNING

We generate two sets of datasets in this case, varying the material microstructure in the following two ways.

**Varying fiber orientation distribution.**   We generate the samples by controlling the predefined parameters of Gaussian Random Field (GRF). With the GRF sharpened by the thresholding, the values to binary field represent two distinct fiber orientations. The binary field is smoothed using a windowed convolution. To address boundary conditions, the matrix is padded with replicated edge values, ensuring that the convolution works uniformly across the entire grid. After the smoothing process, the padded sections are removed, and the remaining field is used to construct the fiber field. We use two fixed fiber angles $\frac{\pi}{3}, \frac{2\pi}{3}$ for the corresponding binary field. We generate 300 material sets, each with 500 loading/displacement pairs, and divide these into training, validation, and test sets in a 200/50/50 split. Exemplar Samples from this dataset are illustrated in Figure 11. These samples demonstrate the variability of $\boldsymbol{b}$, which is of critical for the latent variable identifiability in unsupervised learning settings, as proved in Theorem 2.

**Varying fiber orientation magnitude and segmentation line rotation.**   Instead of controlling the fiber orientation angles on the binary field as two constant values, we sample the orientation distribution consisting of two segments with orientations $\alpha_1$ and $\alpha_2$ on each side, respectively, separated by a line passing through the center. The values of $\alpha_1$ and $\alpha_2$ are independently sampled from a uniform distribution over $[0, 2\pi]$, and the centerline's rotation is sampled from $[0, \pi]$. We generate 300 material sets, each with 500 loading/displacement pairs, and divide these into training, validation, and test sets in a 200/50/50 split.

**Loading and displacement pairs for one microstructure.**   After generating the specimens with varying fiber orientations, we feed the $\boldsymbol{b}^\eta(x)$ as the $\alpha(x)$ to the HGO model, and keep the material property set $\{E, \nu, k_1, k_2\}$ as constant. For each microstructure, we randomly sample loading and displacement pairs for each microstructure from the previous step.

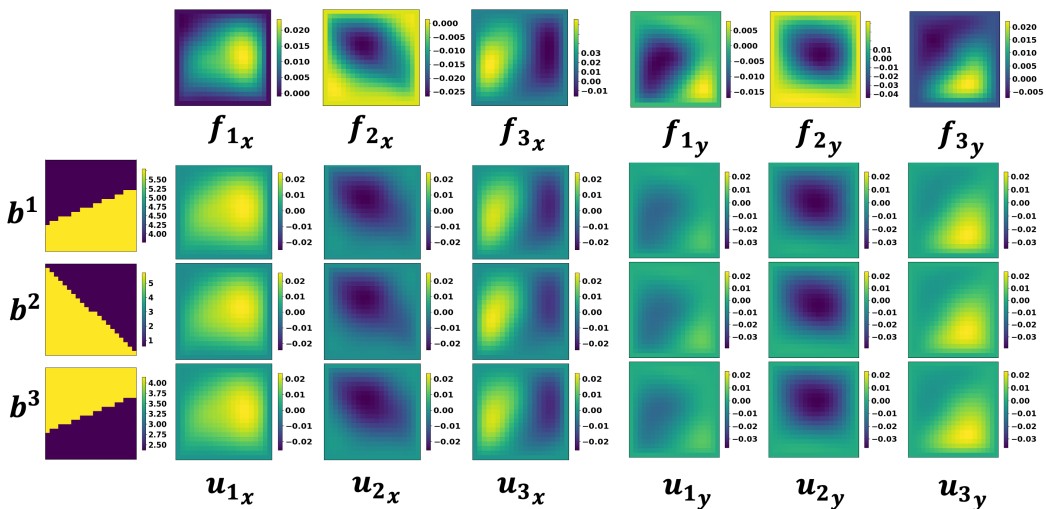

Figure 11: Illustration of fiber orientation magnitude and segmentation line rotation. Upper: three different loading instances of $\boldsymbol{f}_i$. Bottom left: three different microstructure instances of $\boldsymbol{b}^\eta$. Bottom right: corresponding solution fields $\boldsymbol{u}_i^\eta$.

The loading in this example is taken as the body load, $\boldsymbol{f}(\boldsymbol{x})$. Each instance is generated as the restriction of a 2D random field, $\phi(\boldsymbol{x}) = \mathcal{F}^{-1}(\gamma^{1/2}\mathcal{F}(\Gamma))(\boldsymbol{x})$. Here, $\Gamma(\boldsymbol{x})$ is a Gaussian white noise random field on $\mathbb{R}^2$, $\gamma = (w_1^2 + w_2^2)^{-\frac{5}{4}}$ represents a correlation function, $w_1$ and $w_2$ are the wave numbers on $x$ and $y$ directions, respectively, and $\mathcal{F}$ and $\mathcal{F}^{-1}$ denote the Fourier transform and its inverse, respectively. This random field is anticipated to have a zero mean and covariance operator $C = (-\Delta)^{-2.5}$, with $\Delta$ being the Laplacian with periodic boundary condition on $[0,2]^2$, and we then further restrict it to $\Omega$. For the detailed implementation of Gaussian random field sample generation, we refer interested readers to Lang & Potthoff (2011a). Then, for each sampled loading $\boldsymbol{f}_i(\boldsymbol{x})$ and microstructure field $\boldsymbol{b}^\eta(\boldsymbol{x})$, we solve for the displacement field $\boldsymbol{u}_i^\eta(\boldsymbol{x})$ on the entire domain.

## C.4    EXPERIMENT 4 - SYNTHETIC DATASET FOR IDENTIFIABILITY DEMONSTRATION

We demonstrate using experiment 4 that the disentangled latent factors from DisentangO correspond to the true generative factors. We generate synthetic data following the process in equation C.2. We work with latent variables $\boldsymbol{z}$ of dimension 2 and sample from $\boldsymbol{z} \sim \mathcal{N}(\mu_{\boldsymbol{b}}, \sigma_{\boldsymbol{b}}^2 \mathbf{I}) \in \mathbb{R}^2$ where for each microstructure $\boldsymbol{b}$, we sample $\mu_{\boldsymbol{b}} \sim \mathcal{U}[-4,4]$ and $\sigma_{\boldsymbol{b}} \sim \mathcal{U}[1,10]$ and $f \sim \mathcal{U}[0,1] \in \mathbb{R}^3$, $M \sim \mathcal{U}[0,1] \in \mathbb{R}^{2\times2}$, $A = [I, 2I, 3I, 4I]^T$, $M \in \mathbb{R}^{8\times2}$, $b \sim \mathcal{U}[0,1] \in \mathbb{R}^8$, $W \sim \mathcal{U}[0,1] \in \mathbb{R}^{4\times1}$, $l \sim \mathcal{U}[0,1] \in \mathbb{R}$,

$$\theta_P = Az \,, \tag{C.2}$$
$$u = W\sigma(\theta_{P1}f + \theta_{P2}) + l \,. \tag{C.3}$$

We generate 700 data pairs of $(z, \theta_P)$ corresponding to 700 tasks, and for each task, we generate 200 samples of $(f, u)$ pairs.

| $d$ | 3 | 5 | 7 | 9 |
|---|---|---|---|---|
| MCC | 0.7836 | 0.8013 | 0.8237 | 0.9121 |
| $R^2$ | 0.5673 | 0.6026 | 0.6473 | 0.8242 |
| Rank($w$) | 3 | 4 | 4 | 4 |

Table 4: The MCC and $R^2$ scores to evaluate identifiability on the synthetic dataset in experiment 4.

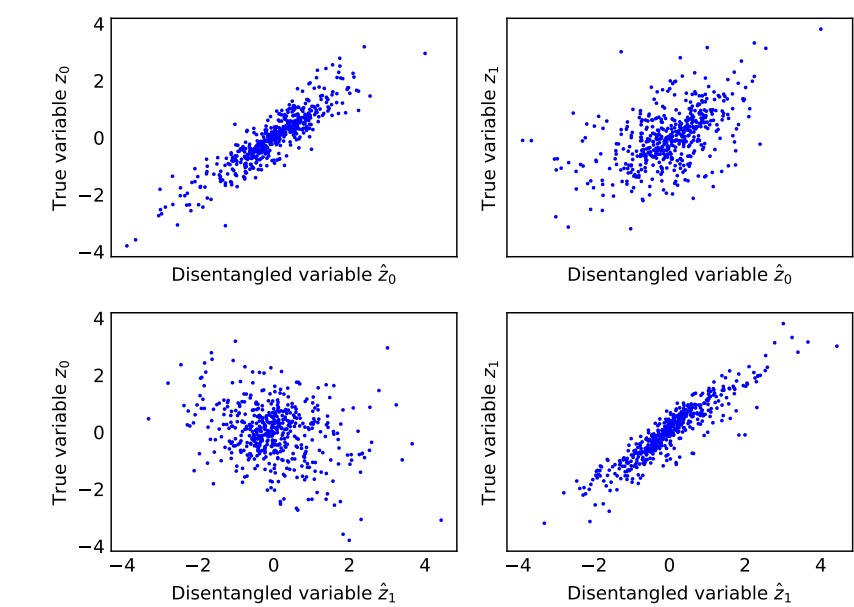

Figure 12: The scatter plot of the true generative variables $z$ and the disentangled latent variables $\hat{z}$ from the synthetic dataset with $d = 9$ in experiment 4.

We conduct experiments on the synthetic dataset and split the tasks into $d/100/500$ for training, validation and test, respectively, with $d$ chosen from the set of $(3, 5, 7, 9)$ to investigate the effect of the number of training tasks on the identifiability. In particular, we measure the component-wise identifiability of the latent variables $\boldsymbol{z}$ by computing the Mean Correlation Coefficient (MCC) and the coefficient of determination $R^2$ scores on the test dataset, and report the results in Table 4, where we observe a monotonic growth on both MCC and $R^2$ with the increase of the number of training tasks $d$. Figure 12 illustrates the alignment on the test dataset between the true generative factors $z$ and the disentangled factors $\hat{z}$ from DisentangO in the case of $d = 9$. To verify Assumption 4, we calculate the rank of the matrix $w(\boldsymbol{z}, \boldsymbol{b})$. Specifically, for each $\theta_p$, with the VAE encoder, we calculate the $\mu_{\boldsymbol{b}}$ and $\sigma_{\boldsymbol{b}}$, sample $z$, and calculate the corresponding $\dfrac{\partial q_i(z_i, \boldsymbol{b}^j)}{\partial z_i}$ and $\dfrac{\partial^2 q_i(z_i, \boldsymbol{b}^j)}{\partial z_i^2}$, where $i = 1, \cdots d_z, j = 1 \cdots d$.

