# OpenReview forum: "Disentangled Representation Learning for Parametric Partial Differential Equations"
_ICLR.cc/2025/Conference — Submitted to ICLR 2025_

### Official Review · Reviewer_gJXj · 2024-10-18

**Soundness:** 2
**Presentation:** 3
**Contribution:** 2
**Rating:** 8
**Confidence:** 3

**Summary:**

The authors introduce DisentangO, a novel architecture that disentangles latent physical factors from neural operator parameters. The model is capable of solving forward and inverse problems at the same time. By combining a multi-task model with a VAE-based approach, DisentangO improves physical interpretability and generalization across PDE-driven systems. The paper offers theoretical guarantees to support the effectiveness of their method.

**Strengths:**

1) The authors present a novel approach based on VAEs that addresses both forward and inverse problems simultaneously.

2) They offer theoretical guarantees to support the effectiveness of their method.

3) The approach is evaluated against MetaNO and basic VAEs.

4) The paper is well-written and mathematically rigorous.

**Weaknesses:**

(1) The work has the potential to be highly impactful. However, several papers that take a similar approach are not cited in this work [1][2][3]. I believe these papers should be acknowledged and their contributions recognized.

(2) The networks used in the benchmarks are relatively small. For example, the FNO architecture that you used in supervised settings has only 8 modes and a width of 26, which is quite modest.

(3) The architecture used for baseline VAEs' encoders and decoders are simple MLPs. I believe that more appropriate and strnoger baselines would be convolutional VAEs, for example.

(4) The model is not compared to the state-of-the-art approaches such as [1][2][3]. The work done in [1][3] could be easily adapted to experiments in the paper.

----

[1] Lingsch, L. E., Grund, D., Mishra, S., & Kissas, G. (2024). FUSE: Fast Unified Simulation and Estimation for PDEs. arXiv preprint arXiv:2405.14558.

[2] Fotiadis, S., Valencia, M. L., Hu, S., Garasto, S., Cantwell, C. D., & Bharath, A. A. (2023, July). Disentangled generative models for robust prediction of system dynamics. In International Conference on Machine Learning (pp. 10222-10248). PMLR.

[3] Tong, G. G., Long, C. A. S., & Schiavazzi, D. E. (2024). InVAErt networks: A data-driven framework for model synthesis and identifiability analysis. Computer Methods in Applied Mechanics and Engineering, 423, 116846.

**Questions:**

1) Why do you use simple MLPs for VAE architectures as baselines and not more advanced architectures?

2) Why are relatively small models, like the 0.7M FNO, being used?

---

> ### Author Response · Authors · 2024-11-25
> **Response to reviewer gJXj, part I**
>
> We thank the reviewer for the insightful comments and suggestions. Our response:
>
> **Additional references**: We greatly appreciate the reviewer for pointing us to these relevant work. We have added these references to the discussion of disentangled representation learning in Section 2 and added FUSE and InVAErt as additional baselines. As FUSE assumes a constant loading field and does not apply to our setting of multiple loading instances, we have also modified the FUSE implementation and created a FUSE extension (denoted as ``FUSE-f'') to allow for solutions from different loading instances as input. The detailed settings are documented in Appendix B2 in the revised paper.
>
>
> **Relatively small models of FNO**: Our datasets have spatial discretizations of 41$\times$41, 29$\times$29, and 29$\times$29 in the three considered experiments, respectively, which correspond to a total of 20, 14, and 14 Fourier modes in maximum, respectively. DisentangO employs implicit Fourier neural operators (IFNO) [1], a provably universal approximator for PDE solvers, as the forward solver backbone. In IFNO, all iterative layers share the same trainable parameters. Hence, the resulting total number of trainable parameters does not increase with the increase of Fourier layers. This makes IFNO more parameter efficient. In our experiments, we keep 8 modes with width 32, which is expected to cover the major modes, leading to the final model of size 0.7M in the first experiment. As this relatively small model already yields satisfactory forward prediction performance in that DisentangO's forward prediction error is only 1.65\%, we did not pursue further increasing the model size. For fair comparison, we adjust the FNO settings to keep the total number of parameters on the same level as DisentangO. As it is unfair to FNO to further truncate the Fourier modes, we keep the number of modes the same (i.e., 8) and reduce the width to match the total number of parameters, which results in the final FNO width of 26.
>
> [1] You, Huaiqian, Quinn Zhang, Colton J. Ross, Chung-Hao Lee, and Yue Yu. "Learning deep implicit Fourier neural operators (IFNOs) with applications to heterogeneous material modeling." Computer Methods in Applied Mechanics and Engineering 398 (2022): 115296.

---

> ### Author Response · Authors · 2024-11-25
> **Response to reviewer gJXj, part II**
>
> **Additional baselines**: We have added four additional baselines to experiment 1, i.e., (1) InVAErt [1], (2) FUSE [2], (3) FUSE-f (a FUSE extension that allows instances from different loading fields as input), and (4) a convolution-based VAE, and report the results in Table 1 in the revised manuscript (also shown below for your convenience).
>
> We point out that the settings of FUSE and InVAErt are actually different from ours, as FUSE assumes constant initial/boundary conditions and do not allow solutions from different loading fields as different samples. Hence, it cannot act as a solution operator for parametric PDEs with multiple possible input functions. On the contrary, in our setting, we consider the solution operator which can handle many different loading instances, and can be used to predict the resulting displacement field for new and unseen loading instances. Because of this difference, we can not use the same training set in the vanilla FUSE and InVAErt. Instead, we only consider a single instance of the loading field and its corresponding solution field for each task to train FUSE and InVAErt. Due to this limitation, FUSE and InVAErt actually use less information compared to DisentangO. As a result, DisentangO outperforms both FUSE and InVAErt.
>
> To be able to use all the instances of different loading fields, we created an extension of FUSE (denoted as ``FUSE-f'') that takes a concatenation of both the displacement field and the loading field as input to the inverse model. For the forward model in FUSE-f, we concatenate the loading field to the output of the first band-limited lifting layer and subsequently use a one-layer MLP to map it back to the original dimension. However, DisentangO still outperforms this setting.
>
> Per the reviewer's suggestion, we have also added an additional baseline by employing the CNN-based architecture in VAEs. As can be seen in Table 1, employing convolution-based architecture did not improve the performance of VAE.
>
> Test errors and the number of trainable parameters in experiment 1. Bold number highlights the best method.
> |Models | #param | Test errors of data | Test error of latent z |
> | :------------- | :-----------: | :-----------: | :-----------: |
> | DisentangO | 697,383 | 1.65% | **4.63%** |
> | MetaNO | 295,842 | **1.59%** | - |
> | NIO | 709,260 | - | 15.16% |
> | FNO | 698,284 | 2.45% | 14.55% |
> | InVAErt | 706,761 | - | 5.16% |
> | FUSE | 706,151 | - | 4.99% |
> | FUSE-f | 707,271 | 16.33% | 6.19% |
> | VAE | 697,672 | 49.97% | 16.34% |
> | convVAE | 663,551 | 81.11% | 16.27% |
> | $\beta$-VAE | 697,672 | 51.16% | 16.47% |
>
> [1] Tong, Guoxiang Grayson, Carlos A. Sing Long, and Daniele E. Schiavazzi. "InVAErt networks: A data-driven framework for model synthesis and identifiability analysis." Computer Methods in Applied Mechanics and Engineering 423 (2024): 116846.
>
> [2] Lingsch, Levi E., et al. "FUSE: Fast Unified Simulation and Estimation for PDEs." arXiv preprint arXiv:2405.14558 (2024).

---

> > ### Comment · Reviewer_gJXj · 2024-11-26
> >
> > I appreciate the authors' detailed explanations and additional experiments. I am satisfied with the responses. If the paper is accepted, I kindly request that the explanations and additional results be incorporated into the camera-ready version.
> > I will increase my score to 8 (accept).

---

> > > ### Author Response · Authors · 2024-11-26
> > > **Thank you!**
> > >
> > > Thank you for your prompt feedback and willingness to increase the score. We sincerely appreciate your time and valuable suggestions. While the additional results have been already added into the revised manuscript (Table 1 and Appendix C.4), we will incorporate further discussions into the camera-ready version too.

---

### Official Review · Reviewer_8Qzf · 2024-10-31

**Soundness:** 2
**Presentation:** 1
**Contribution:** 2
**Rating:** 3
**Confidence:** 3

**Summary:**

This paper introduces DisentangO, a novel variational hierarchical neural operator designed to solve forward and inverse problems simultaneously. The authors present theoretical results to substantiate the method, which is further evaluated through numerical experiments on various benchmark datasets.

**Strengths:**

- The paper presents an innovative architecture that combines a hierarchical variational autoencoder (VAE) with a Fourier neural operator (FNO) to address inverse problems.
- The authors also provide a theoretical analysis on the component-wise identifiability of the model.
- The proposed method outperforms the baselines in the supervised settings.

**Weaknesses:**

- The presentation lacks clarity.
- The experiments conducted on semi-supervised and unsupervised datasets are insufficient, with a notable absence of comparisons to baseline algorithms.
- The learning process does not adequately incorporate information from the partial differential equations (PDEs).

**Questions:**

1. Presentation
- The overall architecture presented in Figure 1 is not sufficiently informative. The meaning of each arrow in the figure is unclear, as some appear to represent feed-forward computations while others do not.
- Additionally, every notation should be clearly defined upon its first introduction; this paper lacks clarity in this regard.
- What are the first decoder and second decoder in Figure 1?
- How did you computed $\theta_P^{\eta}$ from the input? This is not explained in the manuscript.
- How did you reconstructed $b^\{\eta}$ from $z$ for the semi-supervised and unsupervised settings?

2. Experiments
- The authors conducted experiments on several benchmark datasets. What baseline algorithms, aside from neural operators and variational autoencoders (VAEs), were used for these datasets? Additionally, could you provide the test errors for the traditional methods?

---

> ### Author Response · Authors · 2024-11-25
> **Response to reviewer 8Qzf, part I**
>
> We thank the reviewer for the valuable time and the constructive suggestions. Our response:
>
> **Arrows in Figure 1**: We have modified the caption of Figure 1 and explained the different meanings of different arrows in the revised manuscript. Specifically, black arrows indicate forward computations, red arrow denotes where interpretable physics discovery happens, and dashed arrows mark the sources of loss function computation.
>
> **Notation definition upon first appearance**: We have carefully gone through the manuscript and make sure that notation definitions where missing are added upon first appearance.
>
> **First and second decoders in Figure 1**: We have modified Figure 1 as well as its caption to clearly explain the first and second decoders in DisentangO's architecture. In particular, the first decoder refers to the mapping from $z$ (latent representation of each task) to $\theta_P$ (the task-wise neural operator parameter), as demonstrated in equation (3.12). The second decoder denotes the mapping from the input function, $f$ to the output function $u$, when using the task-wise neural operator parameter $\theta_P$. The formulation was provided in equation (3.13) of the original manuscript.
>
> **How to compute $\theta_P^\eta$ from input**: $\theta_P^\eta$ represent the task-wise lifting-layer parameters, which are essentially the weights and biases of the one-layer MLP used as the lifting layer. As in standard FNO models, the lifting layer is used to transform the input data to a higher-dimensional space. In DisentangO, multi-task learning is realized via employing task-wise lifting layers, i.e., each task has its own lifting layer. These lifting-layer parameters (i.e., $\theta_P^\eta$, in other words, the weights and biases of the MLPs) are assumed to contain meaningful physical information of the PDE system, and the DisentangO's encoder is used to extract these critical physical information from the trainable task-wise lifting layer parameters.
>
> **How to reconstruct $b^\eta$ from $z$ in semi-supervised and unsupervised settings**: As the hidden parameter field $b^\eta$ of the PDE is generally not accessible, especially in the semi-supervised or unsupervised settings, one cannot directly reconstruct $b^\eta$ from the learned latent variables $z$. Through disentangled representation learning, the latent variables $z$ are anticipated to contain critical information of $b^\eta$, but there does not exist a direct and explicit mapping from $z$ to $b^\eta$. As a result, one cannot directly reconstruct $b^\eta$ from $z$. However, there are several tricks one can play with to obtain meaningful interpretations from the learned $z$. On one hand, one can feed a randomly selected input into DisentangO and manually define a desired $z$ in the latent space and perform a forward pass using the trained DisentangO. $b^\eta$ can then be visualized from the output, as demonstrated in Figure 6 in the MMNIST experiment. On the other hand, when the true parameter field is available for some tasks, one can construct a mapping from the learnt latent variable $z^\eta$ to $b^\eta$ on these tasks. With this mapping at hand, one can estimate the corresponding parameter field $b$ for all tasks. This is demonstrated in Figure 5 in the third experiment of the revised paper, where we traverse $z$ and visualize what part of $b$ each dimension of $z$ controls. We have included the above discussion in Appendix B4 in the revised manuscript.

---

> ### Author Response · Authors · 2024-11-25
> **Response to reviewer 8Qzf, part II**
>
> **Baselines**: We have added four additional baselines to experiment 1, i.e., (1) InVAErt [1], (2) FUSE [2], (3) FUSE-f (a FUSE extension that allows solutions from different loading fields as input), and (4) a convolution-based VAE, and the results can be found in the table below (as well as Table 1 in the revised paper). DisentangO stills outperforms all baseline models. As the majority of the baselines require ground-truth generative factors for supervised learning and do not support semi- or un-supervised settings, we can only use MetaNO, VAE and $\beta$-VAE as baselines in experiments 2 and 3. We emphasize that DisentangO is one of the very few models that can handle both forward and inverse PDE learning in all the three learning scenarios (i.e., supervised, semi-supervised, and unsupervised), which explains why experiments 2 and 3 only have three baselines. This represents a strong novelty of DisentangO.
>
> Test errors and the number of trainable parameters in experiment 1. Bold number highlights the best method.
> |Models | #param | Test errors of data | Test error of latent z |
> | :------------- | :-----------: | :-----------: | :-----------: |
> | DisentangO | 697,383 | 1.65% | **4.63%** |
> | MetaNO | 295,842 | **1.59%** | - |
> | NIO | 709,260 | - | 15.16% |
> | FNO | 698,284 | 2.45% | 14.55% |
> | InVAErt | 706,761 | - | 5.16% |
> | FUSE | 706,151 | - | 4.99% |
> | FUSE-f | 707,271 | 16.33% | 6.19% |
> | VAE | 697,672 | 49.97% | 16.34% |
> | convVAE | 663,551 | 81.11% | 16.27% |
> | $\beta$-VAE | 697,672 | 51.16% | 16.47% |
>
> [1] Tong, Guoxiang Grayson, Carlos A. Sing Long, and Daniele E. Schiavazzi. "InVAErt networks: A data-driven framework for model synthesis and identifiability analysis." Computer Methods in Applied Mechanics and Engineering 423 (2024): 116846.
>
> [2] Lingsch, Levi E., et al. "FUSE: Fast Unified Simulation and Estimation for PDEs." arXiv preprint arXiv:2405.14558 (2024).
>
> **Incorporate PDE information**: In many real-world applications (e.g., material modeling from experimental measurements), the underlying physical mechanism as well as the governing PDE are unknown. Hence, our setting considers the scenario where the PDE information is hidden, and we aim to disentangle meaningful physical information purely from data without relying on prior knowledge, thus making it generic and applicable to different scenarios.
>
> We also point out that our framework can be easily extended to the cases where partial PDE information is available. One can either explicitly incorporate this information via adopting a particularly designed neural architecture that hard-codes this information (e.g., clawNO [3] for automatic enforcement of mass conservation) or implicitly as an additional regularizer (e.g., physics-informed neural operators [4]). As our proposed DisentangO architecture is general and is not tied to a specific neural operator model (as noted in the footnote in page 5 of the revised paper), incorporating prior PDE information is straightforward and we do not claim novelty in this regard.
>
> [3] N Liu, Y Fan, X Zeng, M Klöwer, L Zhang, and Y Yu. "Harnessing the Power of Neural Operators with Automatically Encoded Conservation Laws." In ICML 2024.
>
> [4] Li, Zongyi, et al. "Physics-informed neural operator for learning partial differential equations." ACM/JMS Journal of Data Science 1.3 (2024): 1-27.

---

> > ### Comment · Reviewer_8Qzf · 2024-11-26
> > **Thank you for detailed response.**
> >
> > Thank you for your detailed response. I believe the questions regarding clarity have been adequately addressed. However, I have a quick follow-up question about the baselines.
> >
> > I assume you are already aware of this, but I’m curious about how you implemented FNO and NIO for the same problem. Given their fundamentally different nature—NIO learns an operator-to-function mapping, while FNO learns a function-to-function mapping—it would be helpful if you could share more details about the specifics of your implementation.
> >
> > Additionally, I find it unusual that the per-epoch time for NIO is approximately half that of FNO, despite NIO having a larger number of parameters (despite NIO incorporating FNO as a component). Do you have any insights into why this discrepancy might occur?

---

> > > ### Author Response · Authors · 2024-11-26
> > > **Thank you and further response to reviewer 8Qzf**
> > >
> > > We thank the reviewer for their prompt response and further discussions.
> > >
> > > **Details on FNO and NIO implementation**: For NIO, we directly take their github implementations and closely follow the setup of "PDE Solve" example in their paper. In this setting, two neural operators (DeepONet and FNO) are stacked to realize the operator-to-function intuition. The first operator maps multiple solution functions to a set of representations (which can be seen as an analog of eigenfunctions), and the second operator infers the underlying parameter field from the mixed representations. As NIO requires the solution field as input, it cannot be used as a forward solver. Hence, NIO only solves the inverse PDE problem, and it can only be applied to the fully supervised setting (example 1) since it requires data for the parameter field. In NIOs, we use four convolution blocks as the encoder for the branch net and a fully connected neural network with two hidden layers of 256 neurons as the trunk net, with the number of basis functions set to 50. For the FNO part, we use one Fourier layer with width 32 and modes 8, as suggested by the NIO authors.
> > >
> > > The reviewer is correct that FNO is originally designed as a function-to-function mapping. Since FNO is not designed for inverse PDE problems, we consider the inverse optimization procedure following [1], and develop a two-phase process to solve the forward and inverse problems sequentially. In the first phase, we construct the forward mapping from loading field $f$ and the ground-truth material parameter $b$ to the corresponding solution $u$ as: $\mathcal{G}^{FNO} \[f,b;\theta^{FNO}\](x)=u(x).$ This can be seen as an analog of the operator $\mathcal{G}$ in our equation (3.7). Then, with the trained FNO as a surrogate for the forward solution operator, we fix its NN parameters $\theta^{FNO}$, and use it together with gradient-based optimization to solve for the optimal material parameters as an inverse solver. Specifically, given a set of loading/solution data pairs $\{(f_i,u_i)\}_{i=1}^N$, we start from a random guess of the underlying material parameters (typically chosen as the average of all available instances of material parameters for fast convergence),  and minimize the difference between the predicted displacement field from FNO output and the ground-truth one:
> > >
> > > $$b^*=argmin_b \sum_{i=1}^N||u_i-\mathcal{G}^{FNO}\[f_i,b;\theta^{FNO}\]||^2.$$
> > >
> > > As the FNO parameters are fixed, we can back propagate this loss and optimize the input material parameters in an iterative fashion. We perform optimization for 500 iterations and take the final material parameters as the optimized ones. As discussed in [1], this approach works only for the scenarios where $b$ is given and with a finite dimension, hence we can only apply this baseline to our example 1.
> > >
> > > The above discussion, together with detailed setups for all baseline models, are provided in Appendix B.2.
> > >
> > > [1] Lee, Doksoo, Lu Zhang, Yue Yu, and Wei Chen. "Deep Neural Operator Enabled Concurrent Multitask Design for Multifunctional Metamaterials Under Heterogeneous Fields." Advanced Optical Materials (2024): 2303087.
> > >
> > > **Per-epoch runtime**: The reason why FNO requires notably longer per-epoch runtime compared to NIO with similar number of trainable parameters are three-folds. On one hand, the FNO implementation has four Fourier layers, compared to just one Fourier layer appended to the end of the DeepONet architecture in NIO. Taking FFT and inverse FFT operations takes slightly more time than simple feed-forward and convolution operations. On the other hand, as discussed above, FNO requires an additional step to solve the inverse problem after the forward mapping is established. To make it a fair comparison, for the averaged per-epoch runtime we report the sum of both the forward and inverse solvers. The averaged per-epoch runtimes for the forward solver and the inverse solver are 6.2 seconds and 2.9 seconds, respectively, accounting for the total per-epoch runtime of 9.1 seconds. Third, NIO takes multiple input functions (in our example it is set to 50) simultaneously while FNO takes one input function at a time. When using the same batch size, NIO actually has a smaller number of batches and therefore trains faster.
> > >
> > > We have added the above details to Appendix B.2.

---

> ### Author Response · Authors · 2024-11-29
> **Awaiting your feedback**
>
> Dear Reviewer 8Qzf,
>
> We deeply appreciate your insightful feedback and thoughtful suggestions on our paper throughout the review and rebuttal process. Your input has been invaluable in helping us refine and strengthen our work.
>
> We are glad to hear that the majority of your concerns have been adequately addressed. To answer your follow-up question on baselines, we have provided sufficient details on the NIO and FNO setup and implementations as well as explanations on the relative per-epoch runtime. These additional details have also been added to the revised paper.
>
> With the discussion period ending soon on Dec. 2, 2024 (Anywhere on Earth), we kindly await your feedback or an updated assessment of our paper. Please let us know if your concerns have been satisfactorily addressed; if so, we would greatly appreciate it if you could update your ratings. We remain available and strive to answer any additional questions.
>
> Thank you once again, and we wish you all the best!
>
> Sincerely,
>
> Authors

---

### Official Review · Reviewer_ASoK · 2024-11-04

**Soundness:** 2
**Presentation:** 2
**Contribution:** 2
**Rating:** 6
**Confidence:** 3

**Summary:**

The authors introduce DisentangO, a novel neural operator architecture based on a variational framing that aims to perform both physics modeling (e.g. forward PDE solving) and governing physics discovery (e.g. inverse PDE solving). The authors provide theoretical analysis showing when the true physical factors can be identified from the learned representations. The effectiveness of the method is demonstrated in supervised, semi-supervised, and unsupervised learning settings. Additionally, the authors demonstrate that the learned VAE embedding space corresponds to physically meaningful properties on the Mechanical MNIST benchmark.

**Strengths:**

- The paper investigates an important question, interpretability in PDE learning, which has been underexplored in the literature.
- The authors provide interesting theoretical results regarding when physical parameters can be uniquely identified.
- The empirical results are promising:
  - Especially in the supervised and semi-supervised settings, the proposed method is shown to match or outperform existing SOTA neural operator models.
  - Furthermore, the distinction of supervised vs. semi-supervised vs. unsupervised learning in the setting of neural operators is interesting, and the proposed architecture is one of the few which can engage with all three regimes.

**Weaknesses:**

- The main weakness of the paper is that the empirical results regarding the claims of interpretability are somewhat limited. The authors do not propose clear metrics for evaluating interpretability:
  - MMNIST: the authors show that the latent space clusters data based on digit (Fig 3, 7, 8) and that the latent space can be interpolated (Fig 6). However, these seem to be standard properties of VAEs, and it's unclear whether meaningful physical interpretability is being achieved.
  - Heterogeneous material learning: the authors show that a DisentangO with a latent dimension of 3 is able to associate a different physically-meaningful characteristic with each of its dimensions. However, this problem is quite toy, and it's unclear whether these insights transfer to larger problems/networks.
- The paper's writing clarity could be improved in places:
  - The proposed architecture is described in the intro/abstract/conclusion as a "hyper-neural operator" and "hierarchical variational autoencoder", but these terms are not clearly defined in the pape. In particular, the term "hierarchical VAE" seems to be used differently from the typical definition in the literature [1], which involves partitioning latent variables into disjoint groups and modeling conditionally ($\prod_l p(z_l \mid z_{< l})$). The model used in the paper seems to be a standard VAE, if I understand correctly.
  - Similarly, the intro mentions that DisentangO learns from "multiple physical systems", when it seems the paper focuses on different instances of the same PDEs/physical systems.
  - In general, the model architecture could be more clearly described in Section 3.1. A clearer caption in Figure 1 could also be helpful.
  - Tables 2 and 3 would be clearer if model names were included in the table rather than the abbreviation number.
- The existing empirical results could be clarified: see questions.

1. NVAE: A Deep Hierarchical Variational Autoencoder. Arash Vahdat, Jan Kautz.

**Questions:**

- Interpretability: is there any way to systematically validate that the latent factors learned by DisentangO correspond to actual physical parameters, especially in the unsupervised setting with large latent dimension?
- In Table 1, it is shown that a MetaNO with half as many parameters as DisentangO matches on forward problem error. How does the comparison look with a parameter-matched MetaNO? Similarly with Table 3.
- How does the choice of neural operator architecture (IFNO for the DisentangO vs. other architectures as baselines) affect the results? How would IFNO compare on the supervised setting, and could the model be improved by using MetaNO's architecture for DisentangO?

---

> ### Author Response · Authors · 2024-11-25
> **Response to reviewer ASoK, part I**
>
> We thank the reviewer for the valuable comments and suggestions. Our response:
>
> **Metrics for evaluating interpretability**: We want to emphasize that metrics for evaluating interpretability can be clearly defined only in cases where the true generative factors are accessible. Our work considers three practical scenarios for using DisentangO, i.e., supervised, semi-supervised, and unsupervised. These scenarios cover the situations whether or not the true generative factors are available and can be explicitly expressed. In the fully supervised setting corresponding to experiment 1, the metric for evaluating interpretability is the latent supervision test error, as the true generative factors are available and the relative error can be computed, as reported in Table 1. In the semi-supervised setting corresponding to experiment 2, the goal of interpreting physics is to discover the underlying microstructure governing the deformation. Under this setting, since the true generative factors are not available, one cannot use any closed-form metric to evaluate interpretability. However, as the generation of the ground-truth microstructure is controlled by the MNIST digits, the disentangled physical information is anticipated to contain such digit information, which will reveal the underlying microstructure as the interpretable physical information. We thus follow the standard evaluation methods in disentangled representation learning and investigate if the learned latent variables indeed contain the digit information via latent traversal, and the results in Figure 6 confirm the physical interpretability of the learned latent variables as they reveal the underlying microstructure information. The third setting of unsupervised learning is analogous to the second setting of semi-supervised learning where the true generative factors are not accessible and can only be evaluated through latent traversal, and the results in Figure 5 (in the revised paper) reveal critical data-generating information such as the border rotation and the fiber orientations of the two segments, which are meaningful in terms of interpreting the working mechanism of the underlying microstructure.
>
> **Applicability to problems with larger dimensions**: We would like to point out that the intrinsic dimension of the parameter field, $b$, is actually not that small. As shown in [1], the intrinsic dimension for the parameter field in example 2 (MMNIST) is actually $>200$. In our approach, we aim to select the most important feature in $b$ for its mechanical responses, and hence $d_z=15$ seems sufficient. Therefore, experiment 2 already serves as a demonstration that the proposed approach can be applied to heterogeneous material applications with large intrinsic dimensions.
>
> [1] Li, Chunyuan, Heerad Farkhoor, Rosanne Liu, and Jason Yosinski. "Measuring the intrinsic dimension of objective landscapes." arXiv preprint arXiv:1804.08838 (2018).
>
> **Hierarchical VAE and hyper neural operator**: We agree with the reviewer that a hypernetwork may be a more appropriate terminology. We have modified the manuscript accordingly, and added a paragraph introducing hypernetworks in Section 2.
>
> **Multiple physical systems**: The same PDEs can be used to describe different physical systems and different physical phenomena by changing its parameters. For example, the Navier--Stokes equation can be used to describe different flow types (e.g., laminar flow and turbulent flow, which are essentially different physical systems that describe different physical phenomena) depending on the values of certain parameters such as the Reynolds number. On the other hand, different instances of the same physical system may refer to different loading fields or different boundary conditions of the same physical system governed by the same set of PDE parameters. We have clarified this in the revised manuscript.
>
> **Further description of the model architecture**: We have followed the reviewer's suggestion and modified both the image in Figure 1 and its caption to better describe the model architecture. Section 3.1 is also revised to match the description in Figure 1.
>
> **Model names in Tables 2 and 3**: Following the reviewer's suggestion, we have modified Tables 2 and 3 and changed the abbreviation numbers to the model names in the revised manuscript.
>
> **Learned latent factor validation**: we have created an additional synthetic dataset and used it to demonstrate that the disentangled latent factors from DisentangO correspond to the true generative factors (please refer to Appendix C.4 in the revised paper). In Figure 12 close to the end of the revised paper, we display the scatter plot between the true generative factors and the disentangled ones, and we can clearly see that they align well on the two diagonal plots, which validates that the disentangled latent factors corresponds well to the true generative factors.

---

> ### Author Response · Authors · 2024-11-25
> **Response to reviewer ASoK, part II**
>
> **Number of parameters in MetaNO**: The number of parameters in MetaNO is chosen such that it matches the number of parameters in the forward solver of DisentangO. The additional parameters in DisentangO is attributed to the hierarchical VAE architecture for disentangling physical parameters for inverse modeling. As MetaNO lacks a mechanism to solve inverse problems, to make a fair comparison on the forward predictability between MetaNO and DisentangO, we match the number of parameters in the forward solvers of the two models.
>
> **Neural operator architectural choice in DisentangO**: As shown in the implicit Fourier neural operator (IFNO) paper, IFNO is able to match or outperform the SOTA NO models in various physical datasets. Besides prediction accuracy, as IFNO significantly reduces the total number of trainable parameters by using layer-independent parameters, we decide to employ IFNO as the DisentangO backbone for the forward solver. Nevertheless, other SOTA NO models such as FNO are equally applicable as the forward solver. We anticipate that this will yield similar forward prediction accuracy. However, standard NO models such as IFNO and FNO cannot handle multiple physical systems and therefore cannot be used to learn the generative factors of variation. In regards to MetaNO, the second decoder of DisentangO shares close structural similarity to MetaNO's decoder, as both models employ shared iterative Fourier layers and projection layers across different physical systems. In this regard, DisentangO already assimilates MetaNO's architecture.

---

> > ### Comment · Reviewer_ASoK · 2024-11-25
> >
> > Thank you for your detailed responses, including the clarifications and additional experiments! I believe the revised manuscript makes the work significantly stronger. I am willing to raise my score accordingly.

---

> > > ### Author Response · Authors · 2024-11-25
> > > **Official Comment by Authors**
> > >
> > > Thank you for your thoughtful feedback and willingness to increase the score in light of the revisions. We greatly appreciate your time and your recognition of the additional work, and hope the changes address your concerns fully. If there are any further suggestions or clarifications needed, please don’t hesitate to let us know.

---

### Official Review · Reviewer_6vb2 · 2024-11-04

**Soundness:** 2
**Presentation:** 2
**Contribution:** 2
**Rating:** 5
**Confidence:** 3

**Summary:**

This paper presents a model based on a neural operator architecture, integrating an encoder-decoder structure to derive interpretable representations and address multiphysics problems by disentangling data into distinct factors.

**Strengths:**

The paper effectively presents a clear motivation and a well-organized introduction.

To the best of my knowledge, the theoretical contribution addressing the identifiability issue is both novel and significant.

**Weaknesses:**

The methodology section would benefit from additional clarification in the following areas.

1. In lines 189 and 204, the term “parameters” referred to both the parameters within the differential equation and those within the neural operator, which may introduce ambiguity. While the authors have distinguished these parameters through notation, we kindly request that the authors revise the manuscript to adopt distinct terminology for each set. This revision would further enhance clarity in distinguishing between the two.

2. In Theorems 1 and 2, separating the assumptions from the main theorem statements could enhance readability. The current formulation may be somewhat challenging to follow, and presenting the assumptions outside each theorem could help clarify their interpretation. For instance, the authors might consider defining all relevant assumptions -- such as density smoothness, invertibility, conditional independence, and linear independence -- at the beginning of Section 3.2. These assumptions could then be referenced as needed throughout the related theorems. Additionally, the authors may wish to discuss the importance and validity of each assumption to support the theoretical framework.

3. The term "latent supervision" appears to denote a semi-supervised learning context (SC2) in the paper. Using consistent terminology throughout the paper would enhance readability and facilitate understanding. If our interpretation is accurate, we kindly request that the authors consistently use "semi-supervised learning when referring to SC2 throughout the paper.

**Questions:**

1. Theorem 1 addresses the identifiability issue up to an invertible transformation, which may not depend on the differential equation parameters. Could the authors clarify the source of this identifiability issue, and specify what guarantees the theorem provides regarding distribution uniqueness despite this transformation? (While Theorem 2 confirms identifiability even without such a transformation, further elaboration could be beneficial.)

2. In the proposed neural operator model, shared parameters are introduced to reduce degrees of freedom. While the architecture appears sufficient to support the presented theorems, could the authors clarify whether this model structure is indeed minimal? For instance, in line 314, the authors parameterize the posterior distribution $q$ with projection weights $\mathcal{P}$, possibly because the weights in $\mathcal{P}$ are fewer than those in $\mathcal{Q}$ or the intermediate layer. Is this reduction the primary rationale behind the model’s design? We would greatly appreciate further details on the trade-offs in using additional weights in $\mathcal{Q}$ or the intermediate layer for parameterization and how this choice impacts the model's overall performance and interpretability.

3. In line 210, the statement that parameters in the equation cannot be learned under zero Dirichlet boundary conditions is intriguing. Could the authors briefly elaborate on this point?

4. Is it possible to empirically validate one of the assumptions in Theorems 1 and 2, such as linear independence, within the simulation? Although these assumptions may be reasonable based on literature in line 302, empirical support could further strengthen the argument.

5. Tables 2 and 3 indicate that increasing the latent dimension improves prediction results. Is there a specific reason, such as computational constraints, for selecting a latent dimension smaller than 30? The authors might consider including experiments with higher latent dimensions to explore this effect further.

6. In Table 1, MetaNO, which is applied solely to forward problems, utilizes fewer parameters than other models. Could the authors clarify the rationale behind this choice? Additionally, could the author provide the inference time or total training time for each model with the specified number of parameters to enable a more thorough comparison?

7. Could the authors briefly explain how parameter classification information was utilized in NIO and FNO within the latent supervision setting in Table 1?

---

> ### Author Response · Authors · 2024-11-25
> **Response to reviewer 6vb2, Part I**
>
> We thank the reviewer for the insightful comments and questions. Our response:
>
> **Adopt distinct terminology for parameters**: We appreciate the reviewer's suggestion. We have modified the manuscript and distinguished the term "parameters" from the differential equations and the neural operators by using "PDE parameters" and "trainable parameters".
>
> **Separate assumptions from theorems**: We thank the reviewer's valuable suggestions. We have moved Assumptions 1-4 to the beginning of Section 3.2 in the revised manuscript, and referenced them in the statements of Theorems 1-2.
>
> Additionally, we have added a paragraph entitled "Discussion on assumptions" at the end of Section 3.2, and extended our discussion in Appendix A.2. Assumptions 1 (Density Smoothness and Positivity) and 2 (Invertibility) are required to guarantee that there exists a smooth and injective mapping $h:=\mathcal{H}\circ r\circ\hat{\mathcal{H}}^{-1}$, mapping from the ground-truth latent embedding $z$ to the learned embedding $\hat{z}$. The smoothness assumption further makes it feasible to take derivatives of $z$ with respect to $\hat{z}$, which supports the permutation-wise identifiability proof for Theorem 2. We note that the smoothness assumption may possibly be relaxed to $C^2$.
>
> Assumptions 3 (Conditional Independence) and 4 (Linear Independence) are needed to show that the Jacobian of $h$ has one and only one non-zero component for each column. Without these assumptions, it is possible that the data from different $\mathbf{b}$ lack variability. This assumption is plausible for many real-world data distributions. For instance, when the prior on the latent variables $p(z|b)$ is conditionally factorial, where each element $z_i$ has a univariate exponential family distribution given conditioning variable $b$:
> $$p(z|b)=\prod_i \dfrac{Q_i(z_i)}{Z_i(b)}\exp \left[\sum_{j=1}^k T_{i,j}(z_i)\lambda_{i,j}(b)\right],$$
> where $Q_i$ is the base measure, $Z_i(b)$ is the normalizing constant, $T_{i,j}$ are the sufficient statistics, and $\lambda_{i,j}$ are the corresponding parameters depending on $b$. This exponential family has universal approximation capabilities. Additionally, we note that this distribution is conditionally independent with
> $$q_i=\log(Q_i(z_i))-\log(Z_i(b))+\left[\sum_{j=1}^k T_{i,j}(z_i)\lambda_{i,j}(b)\right],$$
> and the linear independence indicates that the matrix formed by $$\omega(z,b^\eta)-\omega(z,b^0)=\left(\mathbf{T}'(\lambda(b^\eta)-\lambda(b^0)),\mathbf{T}''(\lambda(b^\eta)-\lambda(b^0))\right),\,\eta=1,\cdots,S$$
> has full rank ($2d_z$). Numerically, with the added synthetic dataset experiment in Appendix C.4, we show that this identifiability can also be largely achieved with Gaussian distributions of distinct means and variances.
>
> **Latent supervision**: We apologize for the confusion raised. The term "latent supervision" in Section 4.1 corresponds to the supervised learning context, as indicated by the subsection title. We have re-ordered the considered scenarios in Section 3.3 to "SC1: Supervised'', "SC2: Semi-supervised'' and "SC3: Unsupervised'' to match the order of the experimental presentation in Section 4. Additionally, We have revised the manuscript and changed "latent supervision" to "latent $z$ (SC1)'' to avoid potential confusion.
>
> **Further elaboration on identifiability in Theorem 1**: The reason why Theorem 1 only guarantees partial identifiability (up to an invertible transformation) is due to the potential lack of PDE parameter variability. Take the extreme case with $d_z=2$ and only 1 PDE parameter $b=[1,1]$ provided, for instance. Consider sources $z_i\sim \mathcal{N}(0,b_i)$, $i=1,2$, then it is a well-known result that any orthogonal transformation of $z$ would have exactly the same distribution. That means, one can transform the latent variable by any orthogonal transformation and cancel that transformation in the encoder, then obtain exactly the same observed data. By adding Assumptions 3-4, it guarantees sufficient variability between physical systems (tasks). As such, we achieve a stronger identifiability (component-wise identifiability) in Theorem 2.

---

> ### Author Response · Authors · 2024-11-25
> **Response to reviewer 6vb2, Part II**
>
> **Posterior distribution parametrization using lifting-layer parameters $\theta_P$**: The multi-task learning architecture in DisentangO closely follows the MetaNO architecture where the lifting layer $\mathcal{P}$ is adopted as the task-wise adaptive layer for meta learning. The MetaNO authors have theoretically proved that a task-wise lifting layer is sufficient to capture the variability in PDE parameter fields,  and also compared its performances with the models using iterative Fourier layers $\mathcal{F}$ and projection layer $\mathcal{Q}$ as the adaptive layer in their paper. It is verified that using the lifting layer as the adaptive layer leads to the best performance both theoretically and numerically. We therefore directly build on top of their findings and use the lifting layer $\mathcal{P}$ for multi-task learning, where the lifting layer parameters $\theta_P$ are assumed to contain the hidden PDE parameter information and used to parameterize the posterior distribution.
>
> **Further elaboration on learning zero Dirichlet boundary conditions**: Consider the form of a discretized linear PDE setting, say a 1D diffusion equation $b(x)\nabla^2 u(x)=f(x)$ with the finite difference method [1]. In this setting, the PDE problem is solved as $K_{b}U=F$, where $K_{b}$ is the so-called stiffness matrix encoding information about the differential operator. $U=[u(x_1),\cdots,u(x_N)]$ and $F=[f(x_1),\cdots,f(x_N)]$ are the vectors of $u$ and $f$ on $N$ uniform discretization points. In this form, imposing the Dirichlet boundary condition $u=0$ on the boundary points $x_1$ and $x_N$ means that the first and last rows of $K_{b}$ are $[1,0,\cdots,0]$ and $[0,\cdots,0,1]$, and all other rows are with the form $[0,\cdots,0,b(x_{i})/\Delta x^2,-2b(x_i)/\Delta x^2,b(x_{i})/\Delta x^2,0,\cdots,0]$. Hence, varying the value of $b$ at $x_1$ and $x_{N}$ does not change the stiffness matrix, and thus has no impact on the PDE solution operator $K_{b}$. That means, $b(x_1)$ and $b(x_N)$ are not identifiable even if the operator $\mathcal{K}_{b}$ is given (unless additional regularity assumption is made on $b$).
>
> [1] LeVeque, Randall J. "Finite difference methods for ordinary and partial differential equations: steady-state and time-dependent problems." Society for Industrial and Applied Mathematics. (2007).
>
> **Empirical validation on theorem assumption**: We agree with the reviewer that an empirical validation would be nice. In the fully supervised case, the conditional independence and linear independence assumptions are automatically guaranteed by picking proper distribution $p(z|b)$ when designing the VAE architecture. In the semi-supervised and unsupervised cases, generally it would be challenging to estimate $p(z|b)$ and validate these two assumptions in practice. To make such a validation possible, one would need at least the true values of $b$ on some tasks. As such, we can infer an empirical distribution of $p(z|b)$ from these tasks.
>
> To investigate such a capability, in the additional synthetic experiment in Appendix C.4, we consider an unsupervised setting and estimate $p(z|b)$ from the trained model. Then, the linear independence condition can be checked by calculating the vector in equation (3.15) for each task $b^\eta$ and forming an $S\times 2d_z$ matrix from all tasks. When the rank of this matrix is $2d_z$, it means that we can select $2d_z+1$ tasks from them with sufficient variability, such that the linear independence condition is satisfied.
>
> We have added the above discussions to Appendix A.2.
>
> **Restricting maximum latent dimension to 30**: The reviewer is correct that the reason for selecting a latent dimension smaller than 30 is due to the increased requirement in computational resource with the increase of latent dimensions. As restricting the latent dimension to 30 already yields satisfactory prediction accuracy (as evidenced by the test errors reported in Tables 2 and 3), we decide to not go beyond a latent dimension of 30. Moreover, restricting the latent dimension also forces the model to focus on disentangling the most important factors of variation in order to maximize the reconstruction accuracy, thus adding benefits in terms of physical interpretability.
>
> **Number of parameters in MetaNO**: The number of parameters in MetaNO is chosen such that it matches the number of parameters in the forward solver of DisentangO. The additional parameters in DisentangO is attributed to the hierarchical VAE architecture for disentangling physical parameters for inverse modeling. As MetaNO lacks a mechanism to solve inverse problems, to make a fair comparison on the forward predictability between MetaNO and DisentangO, we match the number of parameters in the forward solvers of the two models.

---

> ### Author Response · Authors · 2024-11-25
> **Response to reviewer 6vb2, Part III**
>
> **Per-epoch training time for each model**: We have added the per-epoch training time to Table 1, along with four additional baselines (i.e., InVAErt [1], FUSE [2], FUSE-f (a FUSE extension that allows changing loading fields as input), and a convolution-based VAE). Due to the use of Fourier layers, neural operator (NO) based models (such as DisentangO, MetaNO, FNO) are generally more costly than non-FNO based models (e.g., InVAErt, VAE). However, NO-based models benefit from other advantages such as resolution independence. Additionally, owing to the fact that FUSE and InVAErt assume a constant loading field and cannot handle multi-task operator learning, we can only use the displacement solutions from one instance of the total 50 loading scenarios for training. As a result, the total number of training samples drastically decreases and their per-epoch training time is significantly reduced. In contrast, the FUSE-f extension that takes all the loading instances as input has a similar per-epoch runtime compared to the NO-based models. Note also that, despite that InVAErt and FUSE produce relatively accurate results in terms of latent supervision, they require the displacement field as input and therefore cannot predict displacement as output, thus failing to qualify as a forward solver. This explains why the data errors on displacement are not reported for these models.
>
> Test errors and the number of trainable parameters in experiment 1. Bold number highlights the best method.
> |Models | #param | per-epoch time (s) | Test errors of data | Test error of latent z |
> | :------------- | :-----------: | :-----------: | :-----------: | :-----------: |
> | DisentangO | 697,383 | 12.2 | 1.65% | **4.63%** |
> | MetaNO | 295,842 | 9.8 | **1.59%** | - |
> | NIO | 709,260 | 5.6 | - | 15.16% |
> | FNO | 698,284 | 9.1 | 2.45% | 14.55% |
> | InVAErt | 706,761 | 0.1 | - | 5.16% |
> | FUSE | 706,151 | 2.2 | - | 4.99% |
> | FUSE-f | 707,271 | 11.4 | 16.33% | 6.19% |
> | VAE | 697,672 | 2.8 | 49.97% | 16.34% |
> | convVAE | 663,551 | 2.8 | 81.11% | 16.27% |
> | $\beta$-VAE | 697,672 | 2.8 | 51.16% | 16.47% |
>
> [1] Tong, Guoxiang Grayson, Carlos A. Sing Long, and Daniele E. Schiavazzi. "InVAErt networks: A data-driven framework for model synthesis and identifiability analysis." Computer Methods in Applied Mechanics and Engineering 423 (2024): 116846.
>
> [2] Lingsch, Levi E., et al. "FUSE: Fast Unified Simulation and Estimation for PDEs." arXiv preprint arXiv:2405.14558 (2024).
>
> **Results in Table 1**: The results in Table 1 are from the supervised setting in experiment 1 (i.e., Section 4.1). We have clarified the caption in Table 1 to avoid potential confusion. Both NIO and FNO can only handle the fully supervised scenario in SC1 and cannot handle semi-supervised scenarios as in SC2 and unsupervised scenarios as in SC3.

---

> ### Author Response · Authors · 2024-11-29
> **Awaiting your feedback**
>
> Dear Reviewer 6vb2,
>
> We deeply appreciate your insightful feedback and thoughtful suggestions on our paper throughout the review process. Your input has been invaluable in helping us refine and strengthen our work. Following your comments, we have modified our theoretical identifiability analysis along with further elaborations, clarified several important points, generated an additional dataset with additional experiments to validate our identifiability assumptions, and added four additional baselines along with the requested per-epoch runtime to resolve your concerns.
>
> With the discussion period ending soon on Dec. 2, 2024 (Anywhere on Earth), we kindly await your feedback or an updated assessment of our paper. Please let us know if your concerns have been satisfactorily addressed; if so, we would greatly appreciate it if you could update your ratings. We remain available and strive to answer any additional questions.
>
> Thank you once again, and we wish you all the best!
>
> Sincerely,
>
> Authors

---

> ### Comment · Reviewer_6vb2 · 2024-12-03
> **Official Comment by Reviewer 6vb2**
>
> Thank you for your detailed clarification regarding the terminology used for "parameter" and ''latent supervision", as well as the assumptions underlying Theorems 1 and 2.
>
> I now better understand that DisentangO is derived from MetaNO for multi-task learning, with the projection layer being the only component that varies by task. Additionally, your explanation concerning the identifiability issue under zero Dirichlet Boundary conditions, where $\mathcal{b}(x_1)$ and $\mathcal{b}(x_n)$ are treated as boundary values, was highly informative.
>
> If time permits, I would like to kindly request further clarification on the following points:
>
> 1) **Multi-Task Learning Setup**
> Could you provide additional details regarding the multi-task learning setup? For example, in Table 1, could you clarify why this problem is considered multi-task and how displacement solutions from a single instance involve 50 loading scenarios for training (even if this information may already be discussed in the MetaNO paper)? Given that revisions to your manuscript are not possible at this stage, a brief comment or clarification would be greatly appreciated for now.
>
> 2) **Supervised and Semi-Supervised Cases**
> - For the fully supervised case, it appears that DisentangleO directly approximates $\mathbf{b}$ using the loss function described in lines 356-358. Could you provide additional insight into why this model outperforms the standard FNO? Is this improvement attributable to multi-task or meta-learning components of MetaNO?
>
> - For the semi-supervised case, you mentioned that DisentangleO can classify the digits associated with the data, whereas MetaNO cannot (line 419). Could you elaborate on the significance of this classification capability? This seems particularly important given that MetaNO outperforms DisentangleO in solution prediction, as shown in Table 2.
>
> I would greatly appreciate the authors' comments on these points if time allows.

---

> ### Author Response · Authors · 2024-12-03
> **Thanks and additional response to reviewer 6vb2 (Part I)**
>
> We thank the reviewer for their further discussions.
>
> **Multi-Task Learning Setup**: In parametric PDE problems, different (hidden) PDE parameter fields $b^\eta$ lead to different governing PDEs (and correspondingly different tasks), as well as their corresponding forward solvers $\mathcal{G}^\eta$. Hence, varying $b^\eta$ does not change the displacement $u$ or the loading $f$, but the **mapping** between them. In classical neural operators (NOs) such as FNO and DeepONet, one only considers a forward PDE problem and trains an individual NO for each $b^\eta$. However, because all these forward solvers are with a large number of trainable parameters $\theta^\eta$ which are trained individually, it is impossible to identify the key features of $b^\eta$ from these NO parameters. That means, classical NOs cannot solve inverse PDE problems.
>
> We also want to clarify that the displacement solutions involving 50 loading scenarios do not correspond to a single instance, but they correspond to a single task. The PDE parameters $b^\eta$ vary across tasks, which results in different PDE systems. Under each PDE system, we generate 50 instances of $(f, u)$ pairs. In classical neural operators, one aims to learn a separate function-to-function mapping $\mathcal{G}^\eta$ for each task (i.e., each PDE system) $\eta$, using the 50 instances of $(f, u)$ pairs. In our work, we propose to identify the key features of $b^\eta$ by learning from the data corresponding to multiple PDEs simultaneously. To this end, a multi-task learning architecture is considered, and a common model is trained using the data from multiple PDE systems. As such, one can identify the common low-dimensional structure in $\theta^\eta$, which is considered as the key features in $b^\eta$ that provide interpretability.
>
> **Necessity of Multiple $(u,f)$ Pairs for Each Task $b^\eta$**: Intuitively, this is because the underlying PDE is unknown and $b$ can be of high dimensions. As a result, it requires multiple input-output function pairs to determine an operator. Take a simple problem, $b_1u(x)+b_2(x)=f(x)$, for illustration. If given only one function pair $(\tilde{u}(x),\tilde{f}(x))=(\sin(x),\sin(x))$, both $b_1=1$, $b_2(x)=0$ and $b_1=0$, $b_2(x)=\sin(x)$ can be potential solutions for the inverse problem. That means, the inverse problem is not identifiable.
>
> **Comparison with FNO in Fully Supervised Setting**: In this setting, DisentangO outperforms FNO for two reasons. First, the architecture of DisentangO was developed based on IFNO, where all iterative layers share the same parameters. That means, it is generally more parameter-efficient than standard FNO in a deep layer setting. As a result, when with almost the same number of parameters, DisentangO has a lower data error. Second, FNO was not designed to solve the inverse PDE problem, so we have considered a two-phase process to solve the forward and inverse problems sequentially. In the first phase, we construct the forward mapping from loading field $f$ and the ground-truth material parameter $b$ to the corresponding solution $u$ as: $\mathcal{G}^{FNO}\[f,b;\theta^{FNO}\](x)=u(x)$. Then, with the trained FNO as a surrogate for the forward solution operator, we fix its NN parameters $\theta^{FNO}$, and use it together with gradient-based optimization to solve for the optimal material parameter $b$ as an inverse solver. Specifically, given a set of loading/solution data pairs $\{(f_i,u_i)\}_{i=1}^N$, we start from a random guess of the underlying material parameters (typically chosen as the average of all available instances of material parameters for fast convergence),  and minimize the difference between the predicted displacement field from FNO output and the ground-truth one:
>
> $$b^*=argmin_b \sum_{i=1}^N\|u_i-\mathcal{G}^{FNO}\[f_i,b;\theta^{FNO}\]\|^2.$$
>
> Hence, the numerical errors in the forward solution operator would induce errors in the inverse procedure, and (unavoidably) the optimization procedure would also introduce additional errors.

---

> ### Author Response · Authors · 2024-12-03
> **Thanks and additional response to reviewer 6vb2 (Part II)**
>
> **Comparison with MetaNO in Semi-Supervised Setting**: The digit classification provides important information about the underlying microstructure and is used as partial knowledge (i.e., semi-supervision) to disentangle the key features and learn the inverse solver in DisentangO. As MetaNO only solves the forward PDE problem and cannot solve the inverse one, it cannot disentangle the key features from the images/microstructures, and therefore cannot make use of the digit information at all.
>
> We also point out that MetaNO is anticipated to achieve a lower error in solution prediction, since the forward solver of DisentangO is constructed based on MetaNO. The difference is: DisentangO is further equipped with the hypernetwork architecture, which finds the low-dimensional manifold in the NO parameters. While this low-dimensional structure provides interpretability and solves the inverse PDE problem, it serves as an additional constraint when training the NO parameters for each task. Naturally, a minimization problem with an added constraint would result in a larger optimum compared to the unconstrained one. Hence, a smaller latent dimension size results in a stronger constraint, and a larger data error is anticipated.
>
> As the rebuttal period has come to an end, we hope the above discussion further clarifies things and addresses your concerns; if so, we would greatly appreciate it if you could update your ratings. Thank you once again, and we wish you all the best!

---

### Author Response · Authors · 2024-11-25
**Thanks to AC and all reviewers**

We thank the reviewers for the constructive comments and for recognizing the novelty and high impact of our work, the theoretical support of our identifiability analysis, as well as the comprehensive experimental settings. In our response, we have addressed all questions and suggestions raised by the reviewers, and clarified potential misunderstandings. In particular, we have added detailed discussions on the validity of assumptions (see Appendix A.2 in the revised paper), and an additional synthetic dataset quantitatively verifying that the true latent factors can be inferred from DisentangO if these assumptions are satisfied (see Appendix C.4 in the revised paper). We have also added four additional baselines (i.e., two neural operator (NO) baselines of FUSE, FUSE-f and two non-NO baselines of InVAErt and a convolution-based VAE). The proposed DisentangO still outperforms all considered baseline models.

We sincerely appreciate the time and efforts from the AC and all reviewers. Your comments undoubtedly helped us improve the quality of our manuscript, and we have incorporated them into the revised manuscript. We will also release the source codes and the datasets upon paper acceptance to guarantee reproducibility of all experiments. We are more than happy to answer any follow-up questions during the discussion period.

---

### Meta-Review · Area_Chair_3mgW · 2024-12-20

**Metareview:**

This paper proposes a new variational hyper-operator that aims to disentangle physical factors of variation from the neural operator parameters used for solving parametric PDEs. The approach can handle both forward and inverse problems simultaneously. The experimental results demonstrate its effectiveness across various settings, including supervised, semi-supervised, and unsupervised learning.

The reviewer feedback is mixed, with one reviewer giving a score of 8 and another giving a score of 3. During the final discussion, reviewers recognized the relevance of interpretable learning in scientific contexts. While reviewer gJXj strongly advocated for the paper and highlighted its strengths, others indicated that further revisions would be beneficial. In particular, it would be valuable to (i) provide additional examples where the latent embedding offers a clear advantage, and (ii) discuss more thoroughly the practical benefits of interpretability. This includes studying scenarios where it may be preferable to use the interpretable model rather than a black-box alternative. Moreover, there are concerns about the effectiveness of the proposed approach. For instance, how does training and inference times compare for DisentangO and FUSE.

Overall, I found the paper interesting and well-written. Although the authors attempted to address many of the concerns during the rebuttal phase, not all reviewers were convinced. The paper seems borderline in its current state, and I believe another round of revisions to fully address the remaining issues would substantially improve it. Therefore, I recommend rejecting the paper at this time.

**Additional Comments On Reviewer Discussion:**

The authors engaged in discussions with the reviewers. The revised manuscript addressed most of the concerns.

---

### Decision · Program_Chairs · 2025-01-22

Reject